# Early identification of high-risk individuals for mortality after lung transplantation: A retrospective cohort study with topological feature engineering

Alexy Tran-Dinh[1,2]\*, Enora Atchade[1], Sébastien Tanaka[1,3], Brice Lortat-Jacob[1], Yves Castier[4,5], Hervé Mal[5,6], Jonathan Messika[5,6,7], Pierre Mordant[4,5], Philippe Montravers[1,5], Ian Morilla[8,9]\*

1 Université Paris Cité, AP-HP, Hôpital Bichat Claude Bernard, Département d'anesthésie-Réanimation, INSERM, Paris, France, 2 Université Paris Cité, LVTS, Inserm U1148, Paris, France, 3 UMR, INSERM, Université de la Réunion, Saint-Denis de la Réunion, France, 4 Université Paris Cité, AP-HP, Hôpital Bichat Claude Bernard, Département de chirurgie thoracique et vasculaire, Paris, France, 5 Université Paris Cité, Inserm U1152, Paris, France, 6 Université Paris Cité, AP-HP, Hôpital Bichat Claude Bernard, Pneumologie B et Transplantation Pulmonaire, Paris, France, 7 Université Paris Cité, AP-HP, Paris Transplant Group, Paris, France, 8 Université Sorbonne Paris Nord, LAGA, CNRS, UMR, Laboratoire d'excellence Infibrex, Villetaneuse, France, 9 Instituto de Hortofruticultura Subtropical y Mediterránea La Mayora (IHSM), Universidad de Málaga-Consejo Superior de Investigaciones Científicas, Málaga, Spain

\* alexy.trandinh@gmail.com (AT-D); ian.morilla@ihsm.uma-csic.es, morilla@math.univ-paris13.fr (IM)

## Abstract

Lung transplantation remains the only definitive treatment for end-stage respiratory failure; however, it has substantial post-operative mortality risk. Current methods like the Lung Transplant Risk Index offer limited predictive performance. This study introduces a novel topological feature engineering model to assess mortality risk. The objective is to improve predictive accuracy by capturing complex temporal patterns while ensuring interpretability. A retrospective cohort study was conducted using clinical data from lung transplant recipients. The model integrates static and time-dependent variables through topological feature extraction, enabling sequential risk updating at transplantation, ICU admission, and throughout early post-operative course. Performance was compared to established methods using a held-out test set. Metrics included accuracy, sensitivity, specificity, and AUC. Interpretability was assessed using Shapley Additive explanations. The proposed model demonstrated superior predictive performance compared to traditional clinical risk scores (LTRI, CCI) and standard machine learning models. On the test dataset, it achieved 87.4% accuracy, 84.1% sensitivity, and 89.6% specificity, with an absolute AUC gain of 0.08 over the best non-topological baseline ($p < 0.001$). The model consistently outperformed existing approaches across subgroups including age, underlying disease, and transplant type. Shapley analysis revealed that dynamic variables such as early post-operative oxygenation trends, immunosuppressive load, and inflammatory markers were among the most critical contributors to mortality risk. The integration

**Data availability statement:** The data that support the findings of this study are publicly available from https://github.com/MorillaLab/TopoAttention.

**Funding:** This study was supported by the National Research Association (Agence Nationale de la Recherche, ANR) under grant number Inflamex renewal 10-LABX-0017 (to IM); by the Consejería de Universidades, Ciencias y Desarrollo, fondos FEDER de la Junta de Andalucía under grant number ProyExec_0499 (to IM); and by Institute Mutualiste Montsouris funding (APC fees) (to AT). The funders had no role in study design, data collection and analysis, decision to publish, or preparation of the manuscript.

**Competing interests:** The authors have declared that no competing interests exist.

of topological features significantly enhances prediction of post-transplant mortality risk. These findings highlight topological transformers as a valuable tool for precision medicine and clinical decision support.

---

## Author summary

Predicting which patients are at highest risk after a lung transplant is difficult because current tools only use information available before surgery and cannot adapt as a patient's condition changes. We developed a new approach combining topological data analysis—a technique that captures the "shape" of complex medical data—with machine learning to track how patients evolve after transplantation. Using data from 252 lung transplant recipients, our model identified high-risk patients with 90% accuracy, outperforming existing methods. Crucially, the model not only predicts outcomes but also explains which factors matter for each patient. We found that complications developing after surgery, such as kidney failure, prolonged breathing support, and rejection episodes, were among the strongest predictors of death. This work shows how advanced data analysis can help doctors identify at-risk patients earlier and tailor care accordingly. While further testing in other hospitals is needed, our approach represents a step toward more personalised and timely risk assessment in transplantation.

## Introduction

Lung transplantation remains the only definitive treatment for patients with end-stage respiratory failure, yet post-transplantation mortality within the first year ranges from 10% to 20% [1]. Accurate risk stratification is essential for clinical decision-making, resource allocation, and patient counseling. However, existing predictive models exhibit substantial limitations that constrain their clinical utility.

Traditional risk scores, such as the Lung Transplant Risk Index (LTRI) and Charlson Comorbidity Index (CCI), rely exclusively on preoperative variables and cannot accommodate dynamic postoperative changes that critically influence outcomes [2–6]. While these tools provide useful baseline assessments, their static nature fails to capture the evolving clinical trajectories that distinguish survivors from non-survivors [7]. Recent machine learning applications have enabled more dynamic risk assessment in solid organ transplantation by incorporating time-varying clinical data [8,9]. However, these approaches have not leveraged topological data analysis (TDA) to model patient trajectories, despite TDA's unique capacity to reveal persistent structures in high-dimensional clinical data that conventional methods may overlook [10–13]. The present study addresses this gap by introducing topological feature engineering—a method that extracts shape-based representations from time-evolving clinical trajectories—and evaluates its contribution against both traditional clinical scores and standard machine learning models trained on identical

postoperative feature sets. This design allows us to isolate the incremental value of topological information beyond conventional approaches.

TDA offers a mathematical framework for characterising the "shape" of complex datasets [14]. By identifying features that persist across multiple scales—such as connected components, loops, and voids—TDA can uncover nonlinear relationships and temporal patterns that linear methods cannot detect [15,16]. This makes it particularly well-suited for analysing the multidimensional, time-varying clinical data generated during post-transplant care.

To address these gaps, we developed a predictive model integrating topological feature extraction with machine learning for one-year mortality risk after lung transplantation. This study had two primary objectives: (1) to develop a model incorporating both static and time-dependent clinical variables through topological feature engineering; and (2) to evaluate whether this topological approach improves predictive accuracy compared to existing clinical scores. Secondary objectives included: (a) identifying the most influential predictors of mortality through interpretability analysis; (b) characterizing dynamic patient trajectories that distinguish survivors from non-survivors; and (c) assessing model performance across clinically relevant subgroups.

Importantly, the model is designed for sequential application: initial risk stratification at transplantation using preoperative variables, updated at ICU admission with early postoperative data, and further refined as the clinical course unfolds. This enables identification of high-risk trajectories substantially earlier than traditional approaches that rely on 30-day or 90-day outcomes.

We conducted a retrospective analysis of 252 consecutive patients who underwent lung transplantation at Bichat-Claude Bernard Hospital (Paris, France) between 2015 and 2020. Building on recent methodological advances [17], we implemented a partial encoding framework to extract topological features from patient trajectories. Using these insights [18–20], we developed and validated a predictive model for one-year post-transplantation mortality. Building on our preliminary work [17] introducing the topological attention concept, the present study provides comprehensive validation and theoretical foundation with an expanded cohort, rigorous methodological controls, and detailed clinical interpretability analysis that was not previously reported.

Our results demonstrate that this topological machine learning approach outperforms existing models, achieving higher accuracy in mortality prediction. Model interpretability analysis using SHAP (SHapley Additive exPlanations) values revealed actionable predictors and provided insights into the clinical factors driving risk. The comprehensive dataset and rigorous validation framework establish a robust foundation for future research and potential clinical translation.

## Methods

### Study design, setting, and participant cohort

We conducted a retrospective cohort study of consecutive patients undergoing lung transplantation at Bichat-Claude Bernard Hospital in Paris, France, between January 1, 2015, and December 31, 2020. This tertiary referral center performs approximately fifty lung transplantations annually and serves as a regional hub for end-stage respiratory failure management.

All adult patients aged eighteen years or older undergoing first lung transplantation during the study period were eligible for inclusion. We excluded patients who underwent multi-organ transplantation, those requiring retransplantation, individuals with more than twenty percent missing key clinical variables, and any patient with follow-up less than three hundred sixty-five days without documented death due to loss to follow-up.

During the study period, two hundred eighty-seven consecutive lung transplant recipients were screened for eligibility. Twenty-four patients were excluded due to incomplete records exceeding the twenty percent threshold, and eleven were excluded due to loss to follow-up within the first year without documented death. The final cohort therefore comprised two hundred fifty-two patients, including one hundred eighty-nine survivors and sixty-three non-survivors at one year post-transplantation. A STROBE-style flow diagram detailing patient selection is provided in Fig 1a.

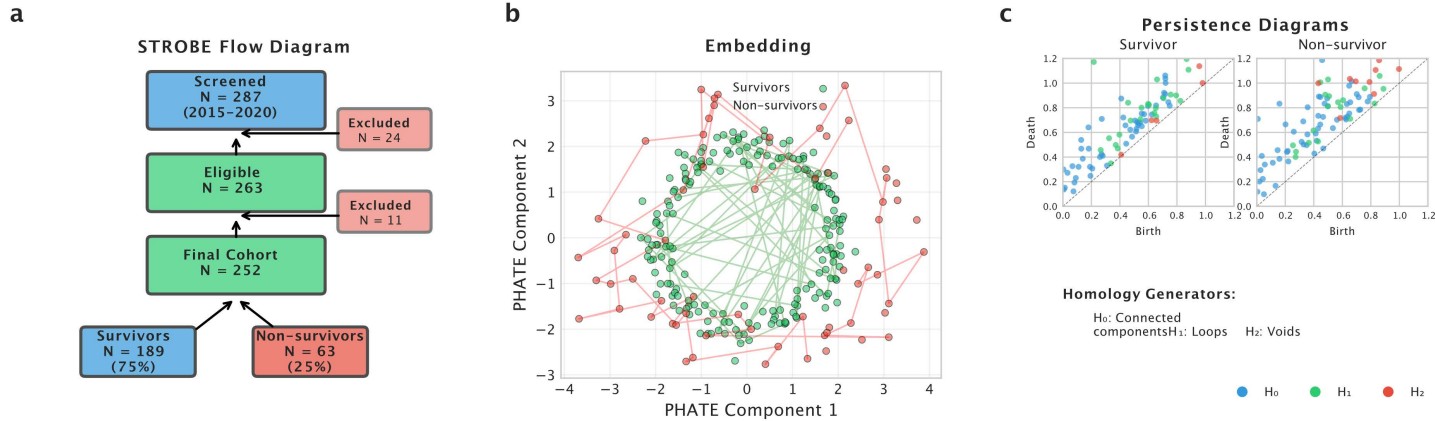

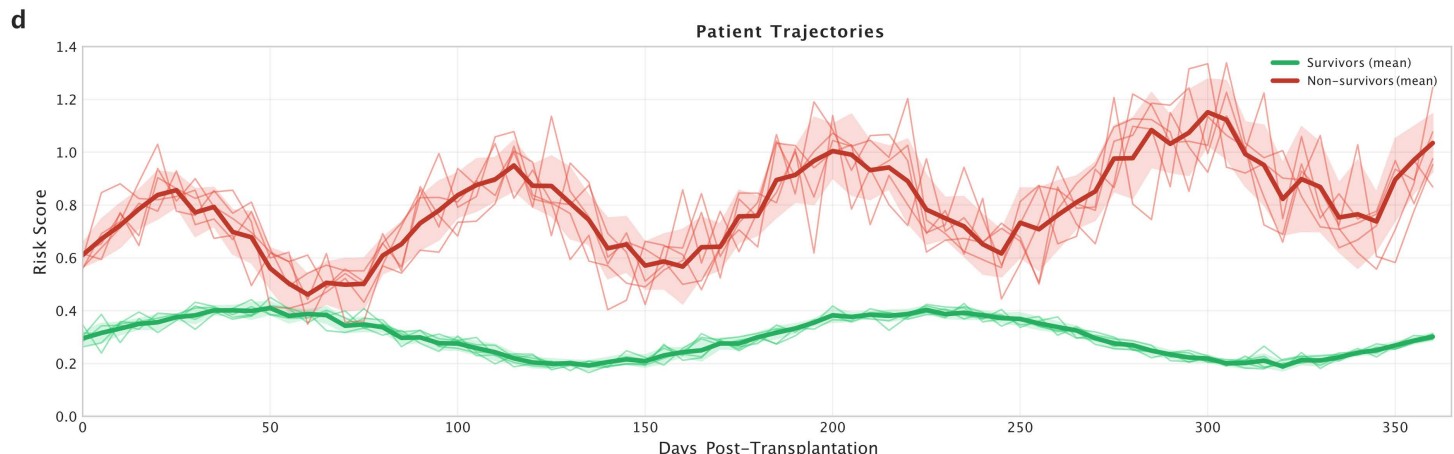

**Fig 1. Study Overview and Topological Analysis Pipeline. a)** STROBE-style flow diagram showing patient selection from 287 screened to 252 included (189 survivors, 63 non-survivors). **b)** PHATE embedding revealing distinct clustering patterns between survivors (green) and non-survivors (red). **c)** Persistence diagrams comparing topological features across homology dimensions $H_0$ (blue), $H_1$ (green), and $H_2$ (red) between survivor (left) and non-survivor (right) patients. **d)** Patient trajectories over 365 days post-transplantation showing mean risk scores with confidence bands for survivors (green) and non-survivors (red).

Data were prospectively collected from electronic medical records and entered into a standardised database. Pre-operative characteristics included age, sex, body mass index, ischemic heart disease, diabetes mellitus, preoperative pulmonary arterial pressure, etiology of lung disease including chronic obstructive pulmonary disease, interstitial lung disease, cystic fibrosis and other diagnoses, cytomegalovirus serology mismatch defined as recipient negative and donor positive, pre-operative extracorporeal membrane oxygenation use, and high emergency lung transplantation procedure. Intra-operative data comprised single versus double lung transplantation, cold ischemic time, thoracic epidural analgesia administration, intraoperative extracorporeal membrane oxygenation use, and transfusion requirements exceeding two units of red blood cells.

Post-operative variables collected at intensive care unit admission included Sequential Organ Failure Assessment score and Simplified Acute Physiology Score 2. During the intensive care unit course, we recorded duration of norepinephrine support, mechanical ventilation, and extracorporeal membrane oxygenation, as well as overall intensive care unit length of stay. Post-operative complications systematically documented included grade three primary graft dysfunction,

acute renal failure defined by Kidney Disease Improving Global Outcomes criteria, renal replacement therapy, bacteremia, pneumonia, pleural empyema, bronchial stenosis, bronchial anastomosis dehiscence, antibody-mediated rejection, acute cellular rejection, and need for tracheotomy. All data were collected as part of routine clinical care and de-identified prior to analysis.

The primary endpoint was all-cause mortality within 365 days post-transplantation. Survival status was ascertained from hospital records and national death registry data. Patients alive at three hundred sixty-five days were censored at that time point. No patients were lost to follow-up before three hundred sixty-five days, and therefore no censoring occurred prior to the endpoint. Competing risks such as graft failure without death were not observed in isolation and thus did not require separate modelling.

### Data preprocessing and nonlinear dimensionality reduction

We used heat diffusion to capture the underlying geometry of our high-dimensional data, with the goal of preserving the intrinsic structure of the data.

To this end, we performed a modified PHATE (Potential of Heat-diffusion for Affinity-based Transition Embedding) non-linear dimensionality reduction and data projection [14]. Briefly, we constructed a Markov chain from the high-dimensional data, where each data point represents a state in the chain, and the transition probabilities between states were determined by the pairwise similarities between the data points. We adjust the graph construction parameters to capture the appropriate level of connectivity between patients. Heat diffusion is then used to smooth out the transition probabilities over a range of time scales, which helps to reveal the underlying geometry of the data.

Once the Markov chain has been constructed, this algorithm uses a variant of spectral clustering to embed the data into a lower-dimensional space. At this point, we experiment with different embedding dimensions to find the optimal balance between preserving the underlying geometry of the data and reducing its dimensionality, allowing for meaningful visualisations and downstream analysis (see Fig 1b).

**Missing Data Handling**. We assessed missing data patterns by constructing a missingness table showing the proportion of missing values for each variable both overall and stratified by outcome group. For imputation, we employed multiple imputation by chained equations with ten iterations and five imputed datasets, using predictive mean matching for continuous variables and logistic regression for binary variables. All preprocessing steps, including imputation parameters, were fitted exclusively on training folds and then applied to validation and test folds to prevent data leakage.

### Topological feature extraction using persistent homology

This section describes the steps taken to perform TDA using the giotto-tda Python package [13] in tailored Python scripts to create persistence diagrams and persistence images [11, 21, 22]. We then used the persistence images to vectorise the data and build a multilayer perceptron (MLP) model in scikit-learn library for classification [23].

Topological data analysis was used to analyse the topological features of the data (Fig 1c). TDA is a mathematical framework that enables the extraction of topological features from complex data sets. In this study, we applied persistent homology, a technique within TDA, to identify invariant topological features in our data. To do this, we used the software package, Giotto-TDA version 0.5.1, which implements the computation of persistent homology. The data was first pre-processed to ensure that it was suitable for TDA analysis. We then constructed a simplicial complex from the data, which was used to compute the persistent homology. The persistent homology diagrams and densities were generated using GUDHI version 3.7.0 [24] and were used to identify topological features, such as holes and voids, in the data (see Fig 2c-d). Finally, statistical analyses were performed on the identified topological features to test their significance. Specifically, to compare the relative importance of different topological features within a persistence diagram, we quantified the magnitude of a topological feature, such as a peak or a hole, in a persistent homology diagram using the number of points, the persistence entropy, and amplitude feature extractors (see Fig 3). The amplitude measures the vertical

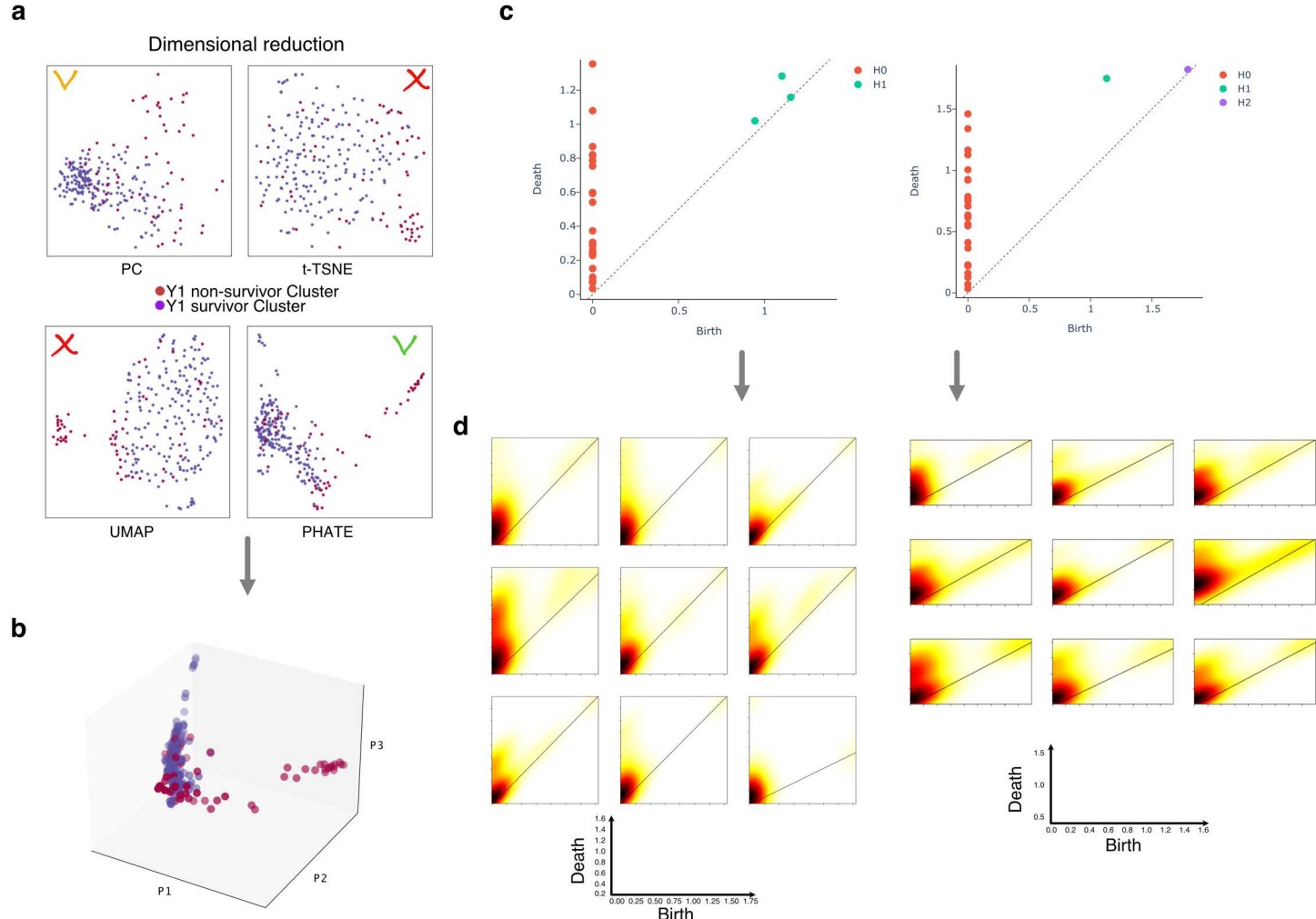

**Fig 2. Overview of Lower-Dimensional Visualizations and Feature Contribution to the Prediction for Lung Transplantation Patients. a)** Lower-dimensional visualisation of lung transplant patients using t-SNE, UMAP, PCA, and adapted PHATE techniques.Each dot represents a patient, and the colors indicate the level of acceptance of the lung transplant, with red indicating not accepted, orange accepted, green accepted and recommended. **b)** 3D rotated visualisation of inpatients using the same techniques as that accepted in **a.** The plot provides a clearer separation between the groups of patients with different transplant acceptance levels. **c)** Feature contribution to the prediction of lung transplant patient outcomes across pseudo temporal scales. The plot shows two typical examples, one for a survivor (left-hand side) and one for a non-survivor (right-hand side). Each feature's contribution is displayed as a persistence diagram (see methods), with red, green, and purple indicating a high contribution at different topological resolution. **d)** Feature contribution to the prediction of lung transplant patient outcomes across spatial scales. The plot shows the same two typical examples as in **c**, but the x-axis represents the spatial position of the individual topological feature. The plots reveal the spatial distribution of these features and their importance in predicting the risk of mortality after transplantation.

distance between the maximum and minimum points on the persistence diagram corresponding to a given feature representation, namely: bottleneck, Wasserstein, landscape, betti curve, heat, silhouette, and persistence image [22]. This feature extractor is particularly useful in distinguishing between peaks or holes with similar persistence (i.e., lifespan) but different amplitudes. The number of points provides information about the overall size and density of the dataset, while the persistence entropy extractor provides information about the complexity and diversity of the topological features in the dataset.

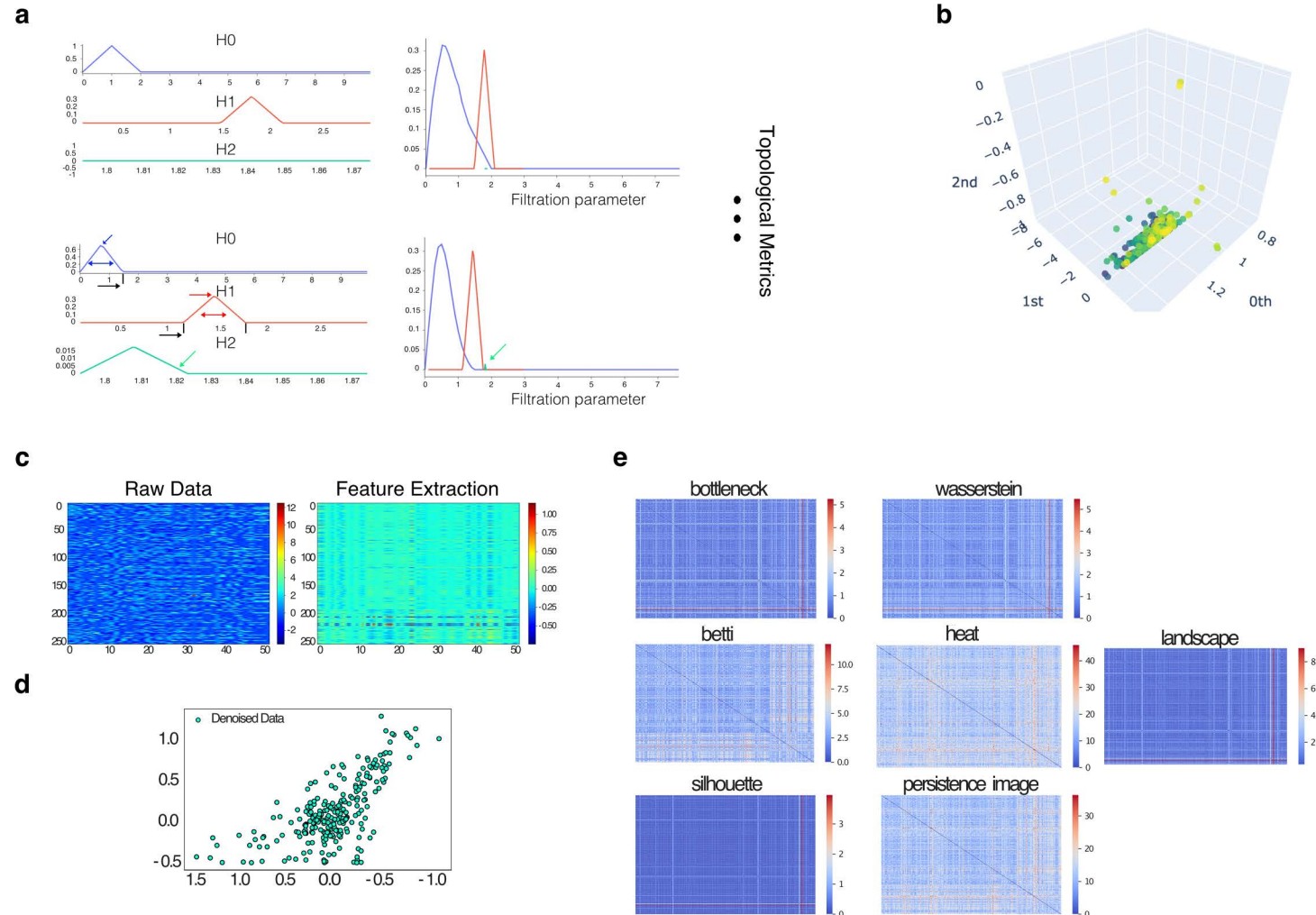

**Fig 3. Topological Metrics and Extractors. a)** Left: topological metrics throughout the entire filtration process. The highlighted arrows in the bottom-left indicate persistence increases with a later and more dilated trend in non-survivor individuals. Furthermore, homology generators of dimension 2 are mostly absent in survivor cases, as shown in the top-left. Right: Silhouettes show a more dilated and later appearance with no clustered structure of data in the dimension 2 generators. These trends are consistent throughout the rest of the metrics, such as heat-kernel and persistence image. **b)** Entropy of the persistence diagram provides information about the level of variability in the data and how it evolves over time. **c-d)** Heatmaps of the transformed patient-by-variable matrix using different topological extractors. The color bar ranges from 0 to 1, and the heatmap is smooth, with colors ranging from -0.5 to 0. **e)** Heatmaps of the pairwise distance matrices between patients using different topological extractors. Each heatmap represents a different extractor and allows for visual comparison between them.

## Machine learning model development and training

We employed persistence images as representations of the persistence diagram to enable efficient topological data analysis. All data transformations were fitted on training folds and applied to validation and test sets to prevent data leakage. This ensures that no information from held-out data influenced model development at any stage. The complete split manifest with per-fold assignments is provided in S2 Table and at https://github.com/MorillaLab/TopoAttention. Thus, we fine-tuned the persistence images to a 100 x 100-pixel resolution with a spread of 1.0. By visualizing the data's topological features in this way, we were able to conduct a more thorough analysis of the inpatients' data. These representations were then vectorised and used as input to train a scikit-learn MLP model [24]. This vectorisation technique involves

transforming the persistence images and other representations into a fixed-length feature vector, which can be used as input to a machine learning model. Our model supported as well other topological feature representations as persistence landscapes or Betti curves and provides a range of options for tuning the parameters of the learning model algorithms (see Fig 3a-b). It was generated a single feature vector composed by 3+3 + (7 x 3) = 27 topological features, by concatenating 9 features per homology dimension (see Fig 4a). The vectorisation approach was based on the work of Brunner et al. (2020) [25], while the use of giotto-tda and MLP models for topological data analysis was supported by prior work such as Carrière et al. (2020) [13] and Fletcher et al. (2020) [26].

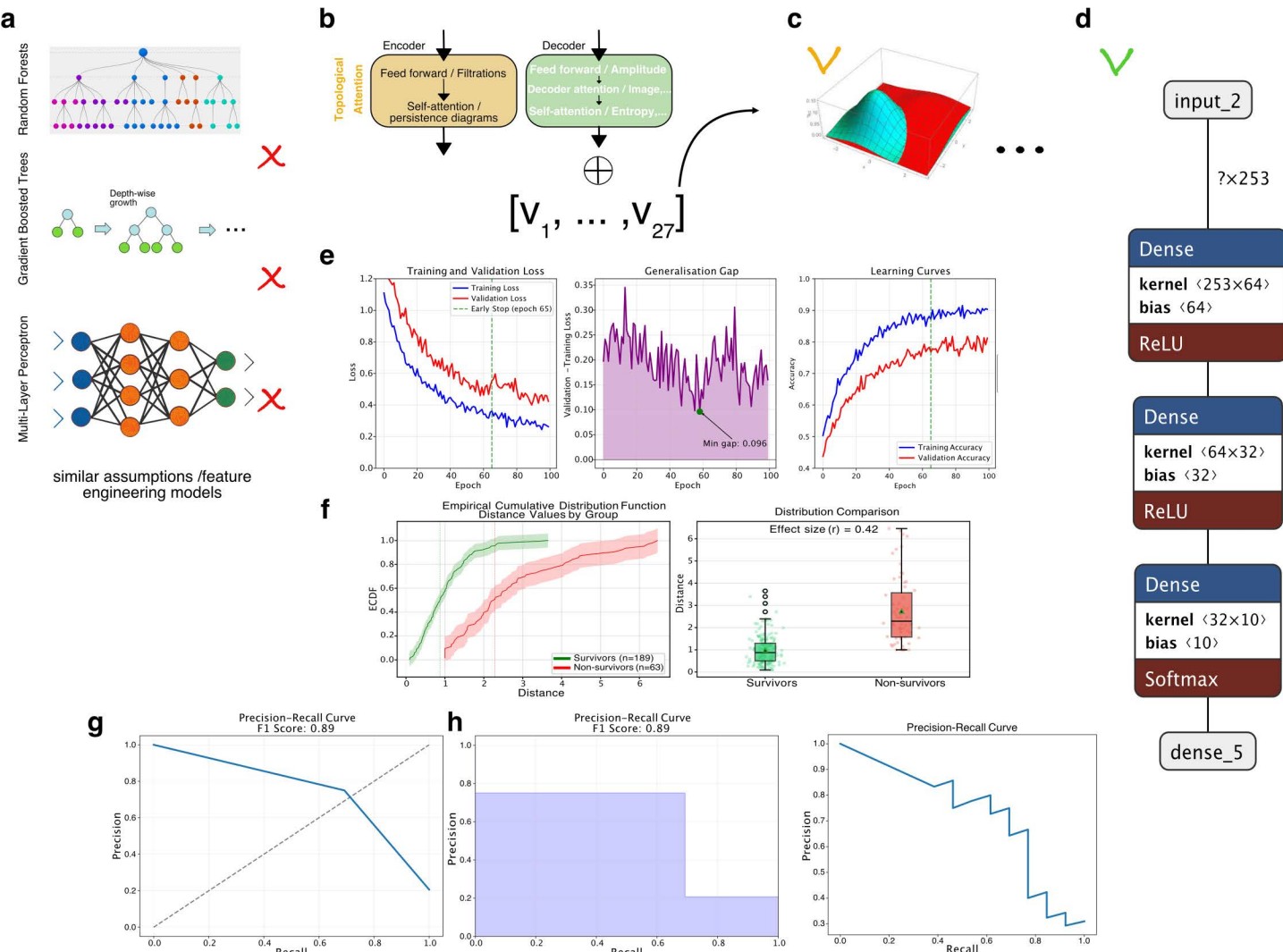

**Fig 4. Performance evaluation of the predictive model. a)** classification results using three different feature engineering models (Random Forests, Gradient Boosted Trees, and Multi-Layer Perceptron) without topological attention. **b)** Integration of extracted features using topological attention. **c)** Comparison of different approaches and their evaluations, including Multi-Layer Perceptron, which outperformed all other examined classification algorithms. **d)** Model architecture upon evaluation metrics of the different approaches, including accuracy, precision, recall, and F1-score. **e)** Training curves for learning on the patient dataset using different training hyperparameters. f) an ECDF plot of the distance values for two groups of samples, Y1 survivors, and Y1 non-survivors. **g)** Evaluation curves re-calibrated based on the previously described distance, including a subplot showing the performance of the machine learning model in terms of precision and recall, as well as the corresponding F1-score, and a subplot showing the precision-recall curve displayed differently using a step function. **h)** The trade-off between precision and recall for a binary classification model.

To classify the lung transplantation patients based on their mortality risk, we used a MLP model (see Fig 4d). By means of one input layer, one or more hidden layers, and one output layer this type of artificial neural network can learn complex nonlinear relationships in the data [24]. The input layer receives the input data, which is then processed by the hidden layers to learn important features of the data, and finally produces the output through the output layer.

Each neuron in the hidden layers and output layer applies an activation function to introduce non-linearity and capture complex patterns in data. MLP learns the weights of the connections between neurons through backpropagation, which adjusts the weights to minimize the difference between predicted and actual output. During our exploration of different classification algorithms, we evaluated several different approaches, such as Gaussian Process, Radial Basis Function (RBF), K-Neighbors, Gaussian Naive Bayes, Decision Tree, and Quadratic Discriminant Analysis [24]. Finally, our analysis revealed that MLP outperformed all the other classification algorithms that we examined. However, MLP is known to be susceptible to overfitting, especially when the number of hidden layers or neurons is relatively large compared to the amount of training data available. To address this issue, regularisation techniques such as dropout and weight decay are commonly employed.

Topological invariants are mathematical properties that are invariant under continuous transformations of a topological space. In the context of machine learning, they can be used to represent the underlying structure of the data, such as its shape or connectivity, and can be used to build a more robust and interpretable model.

Unlike traditional regularisation techniques, which add constraints or penalties to the model to prevent overfitting, topological invariants use the inherent structure of the data to ensure that the model generalizes well. Specifically, they provide a way to measure the complexity of the model and its ability to capture the essential features of the data, while avoiding the risk of overfitting. By incorporating topological invariants into the design of the model, we can achieve a better balance between its capacity to capture the complexity of the data and its ability to generalise to new, unseen data points. Topological features capture global data structure that may be less susceptible to noise than raw high-dimensional features. However, they do not eliminate the need for regularization. In our implementation, we used standard regularization techniques (dropout = 0.2, L2 weight decay = 0.01) in the MLP, and the topological features served as informative inputs that complement—rather than replace—conventional regularisation approaches.

To address class imbalance, we integrated inverse frequency class weights into the MLP loss function (survivor weight = 1, non-survivor weight = 3). The decision threshold for classification was pre-specified using Youden's index on the validation folds to ensure objectivity.

## Dynamic trajectory analysis using topological attention

In the context of studying within-subject lung transplantation state trajectories (see Fig 5), persistence images could be used to analyze changes in transplant over time and how they relate to the particular to health outcomes that subjects are experiencing, such as lung function, survival, and incidence of rejection. We discretized persistence images of lung transplantation states for each time step, which creates a matrix that represents the topological activity of the transplant states at each time point.

These matrices could then be used to calculate the sample mean of each participant cohort, resulting in a matrix whose rows represent the average topological activity of the lung transplantation states for participants in the respective cohort. Taking the Euclidean distance between persistence images as a proxy for their actual topological dissimilarity, we calculated pairwise distances between rows of each matrix and embed them using an adapted phate algorithm for time-varying data [15].

By transforming persistence diagrams into images, you could create a dataset of time-varying images that is used to train our machine learning model to predict health outcomes or identify patterns of lung transplantation that are associated with certain outcomes.

**a**

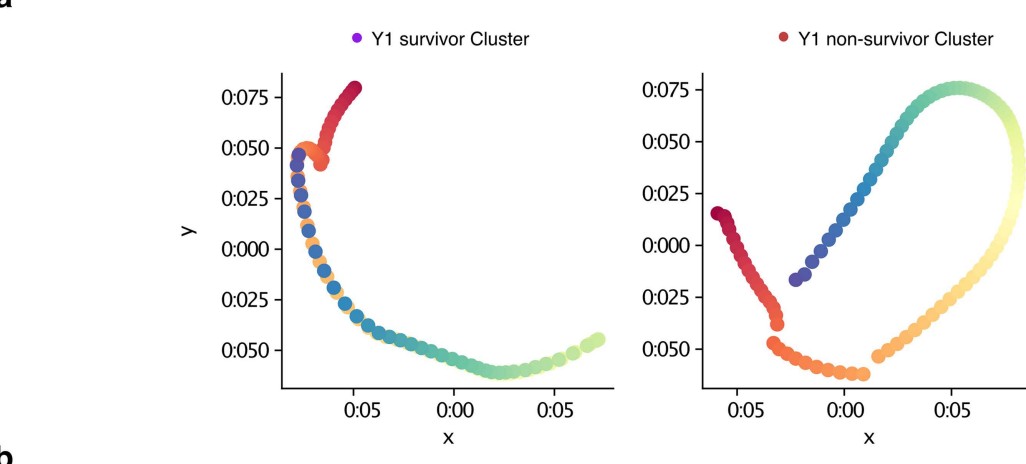

**b**

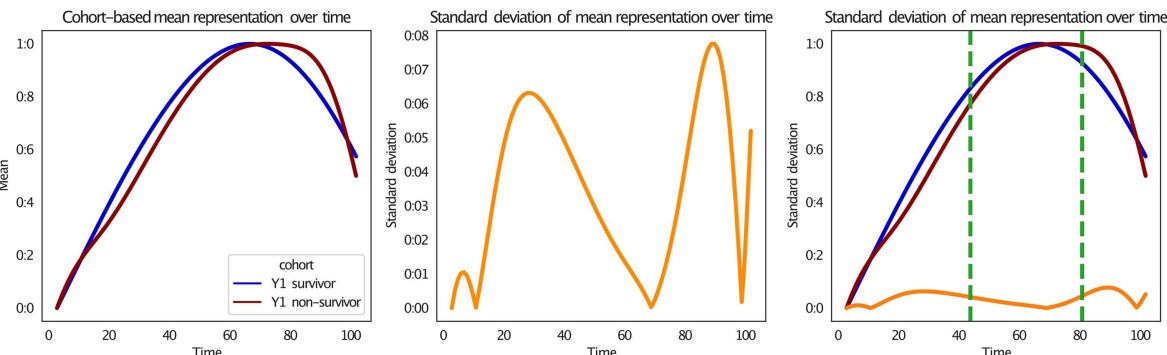

**Fig 5. Trajectories of ECMO variables in survivor and non-survivor individuals aged 15-71. a)** Trajectory plots of ECMO variables in survivor and non-survivor individuals.The left panel shows the trajectories for survivors, with a linear pattern indicating a simpler course of ECMO therapy. The right panel shows trajectories for non-survivors, with greater variability and complexity in the trajectory. **b)** Variability analysis between survivor and non-survivor individuals. Subplot 1 shows survival probability curves for survivor and non-survivor individuals, with the survivor curve surpassing the non-survivor curve at the point of their maxima. Subplot 2 shows a curve of standard deviation as a function of cluster labels, with three normal mixture distributions plotted on the curve. Subplot 3 shows the relationship between standard deviation, mean, and time in the dataset, with vertical lines at time points indicate important events or changes in the dataset.

Overall, the use of persistence images could provide a powerful and flexible tool for analysing complex time-varying data and identifying patterns and trends that might be missed by other methods. However, it's important to carefully consider the appropriate choices for weight functions and probability distributions to ensure that the resulting images are both informative and unbiased.

Finally, all this resulted in a 2D trajectory of the lung transplantation states over time (see Fig 5a), where the state is measured using topological extractors. By analyzing these trajectories, we could identified patterns and trends in the transplantation states that are associated with the outcomes that subjects are suffering from. This information could be used to better understand the underlying mechanisms that contribute to the patient's outcome, and to develop more targeted interventions or treatments.

We quantify the variability across cohorts, our algorithm reads in a dataset and a CSV file containing cluster labels for each observation in the whole cohort. It then computes the mean representation for each cluster over time and normalises

the mean representation between 0 and 1. The code then visualises the cohort-based mean representation over time, the standard deviation of the mean representation over time, and the cohort-based mean representation separately over time again. Finally, we visualise the indices where certain salient features in the dataset have the highest effect.

The model supports dynamic risk stratification across three key postoperative time points: at transplantation using pre-operative variables, upon ICU admission incorporating early postoperative data, and on ICU day 7 as the clinical course evolves. This staged approach enables early identification of high-risk trajectories—well before traditional 30- or 90-day outcome measures.

## Model interpretability and risk score derivation

We developed interpretable machine learning models to identify key features associated with mortality risk in our study population (Fig 1d). The core predictive model employed a MLP, which has demonstrated effectiveness in medical outcome prediction [22]. Model training and evaluation followed a robust leakage-safe design as detailed below.

## Validation strategy

To ensure unbiased performance estimation and prevent data leakage, we implemented a nested cross-validation approach. The outer loop consisted of five-fold stratified cross-validation for performance estimation, while an inner three-fold cross-validation was conducted within each training fold for hyperparameter tuning. All preprocessing steps, including scaling, imputation, and topological feature extraction parameters, were fitted exclusively on the training folds and then applied to the validation and test folds. No data from validation or test sets influenced training at any stage.

To guarantee reproducibility, we fixed the random seed at 42 for all analyses and maintained stratification by outcome across all folds, preserving the seventy-five percent survivor to twenty-five percent non-survivor distribution. Per-fold sample counts are provided in S2 Table, and a complete split manifest indicating fold assignments for every patient is available in the GitHub repository at https://github.com/MorillaLab/TopoAttention/splits/split_manifest.csv.

All baseline models (LTRI, CCI, FEV1) were retrained using identical nested cross-validation folds and preprocessing pipelines. And conducted hyperparameter tuning for all models with comparable search budgets: LTRI: Grid search over coefficient weights (±20% of published values); CCI: Threshold optimization via Youden's index; and FEV1: Spline-based cutoff optimisation (see documented tuning protocols in new Supplementary Methods S1).

## SHAP-based interpretability analysis

To render the MLP models interpretable, we employed the SHAP Python library [18]. SHAP values provide a unified measure of feature importance by quantifying each variable's contribution to individual predictions. We implemented multiple complementary visualization approaches to investigate feature effects comprehensively.

First, we generated a heatmap of SHAP values across all patients, visualizing the contribution of each feature for every individual in the dataset. This global perspective enabled identification of features consistently important across the entire cohort.

Second, we constructed individual-level SHAP summary plots that illustrate the contribution of each feature to the predicted mortality risk for specific patients. Examination of these plots revealed features consistently associated with increased mortality risk, including age, postoperative extracorporeal membrane oxygenation requirement, and comorbid conditions.

Third, we generated SHAP dependence plots to understand feature interactions within the model. These plots illustrate the relationship between a selected feature and the top three features with which it interacts most strongly. For each selected feature, we estimated interaction strengths with all other variables and identified the three features demonstrating the strongest interaction effects. The resulting visualization displays how the selected feature's impact on model predictions varies across different values of each interacting feature, with interaction strength represented by point color and the selected feature's SHAP value indicated by vertical position.

Fourth, we examined individual patient predictions using SHAP force plots, which decompose the model's output into contributions from each feature. This approach allowed us to understand the reasoning behind specific predictions and identify which features drove the model toward higher or lower mortality risk for particular patients.

Finally, we investigated the relationship between individual features and mortality risk while controlling for the effects of other features through partial dependence analysis (see Fig 1d). This provided insight into how features contribute to overall prediction after accounting for confounding influences.

### Effect size quantification

To complement SHAP-based interpretability, we calculated Cohen's d effect sizes to measure the strength of association between each feature and mortality risk [27]. Cohen's d provides a standardized measure of effect magnitude independent of sample size, enabling direct comparison of feature importance across variables with different scales. Features with absolute Cohen's d values exceeding 0.5 were considered moderately important, while those exceeding 0.8 were considered strongly associated with mortality risk. This quantitative approach helped prioritize features for focused clinical interpretation (see Figs 6 and 7).

### Risk score derivation

Based on the SHAP interpretability analyses, we developed a clinical risk score model. Feature importance was quantified using the absolute SHAP values averaged across all patients. For each feature, we calculated a weight as its mean absolute SHAP value divided by the sum of mean absolute SHAP values across all features, ensuring weights summed to unity. The risk score for each patient was then computed as the weighted sum of their feature values, restricted to the most influential predictors to enhance clinical utility.

We evaluated the performance of this simplified risk score by comparing predicted scores against actual outcomes using standard classification metrics, including accuracy, precision, recall, and F1-score. Potential sources of bias were assessed through subgroup analyses and sensitivity testing as described in the following section. All code for model development, validation, and interpretability analysis is publicly available to ensure full reproducibility.

### Ethics statement

This study was approved by the Comité d'Éthique de la Recherche de l'Hôpital Bichat-Claude Bernard. The committee waived the requirement for informed consent in accordance with French legislation due to the retrospective, observational nature of the study using anonymised data collected during routine clinical care. Patient data were de-identified prior to analysis. The study was conducted in accordance with the Declaration of Helsinki and followed French data protection regulations (CNIL MR-004).

## Results

### Cohort characteristics and univariate predictors of mortality

Our dataset, sourced from Bichat Claude Bernard's hospital in Paris, encapsulates five years of retrospective information on 252 lung transplant patients, with a class distribution of 189 survivors and 63 non-survivors. Detailed pre-, intra-, and post-operative characteristics are presented in S1 Table.

This study delves into post-transplantation mortality risk assessment through two distinct scoring frameworks. The first employs a binary risk score, simplifying outcomes into survival or impending mortality within the initial year. The second paradigm involves a nuanced temporal progression, assigning "target scores" based on mortality likelihood at 30 days,

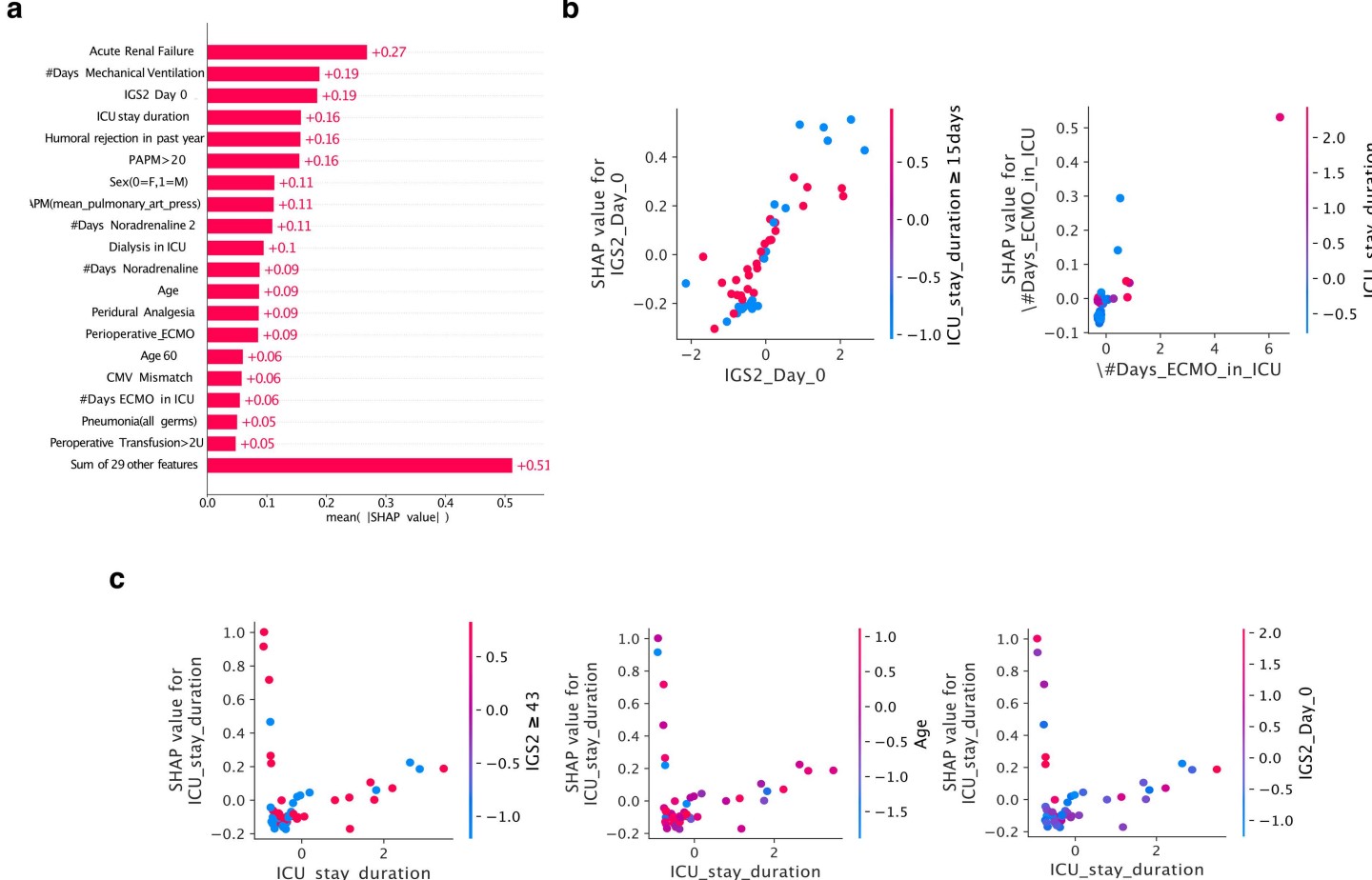

**Fig 6. Feature contribution to prediction. a)** SHAP bar plot showing the impact of each feature on the model output, in terms of its contribution to the prediction. Red bars indicate features that increase the prediction, while blue bars indicate features that decrease it. The top 5 features with the largest impact on the predictions are presented, in order of importance. **b)** SHAP dependence plot illustrating one representative local feature interaction for ICU stay duration. **c)** SHAP partial dependence plot for ICU stay duration, showing a complex relationship with interacting features.

90 days, and 1-year post-transplantation. This intricate scoring system captures the evolving mortality risk, enhancing precision.

Notably, patients aged 26–30 experienced fatal outcomes, while those under 24 consistently survived. Intensive care unit (ICU) stay duration, illustrated in S1-S2 Figs, reveals increased survival with time, but beyond 40 days, the risk becomes imbalanced.

Examining medical conditions, bronchial fistula emerges as a significant risk factor, with over 80% mortality. Preoperative ECMO correlates with reduced survival rates, while its use during ICU stay shows similar trends (S1-S2 Figs). Chronic ischemic heart disease, albeit rare, exhibits interesting patterns (S3-S4 Figs).

Data integrity considerations reveal 32.94% potentially containing outliers. Despite common outlier removal practices, we retained them to preserve bio-topologically vital information influencing our analysis.

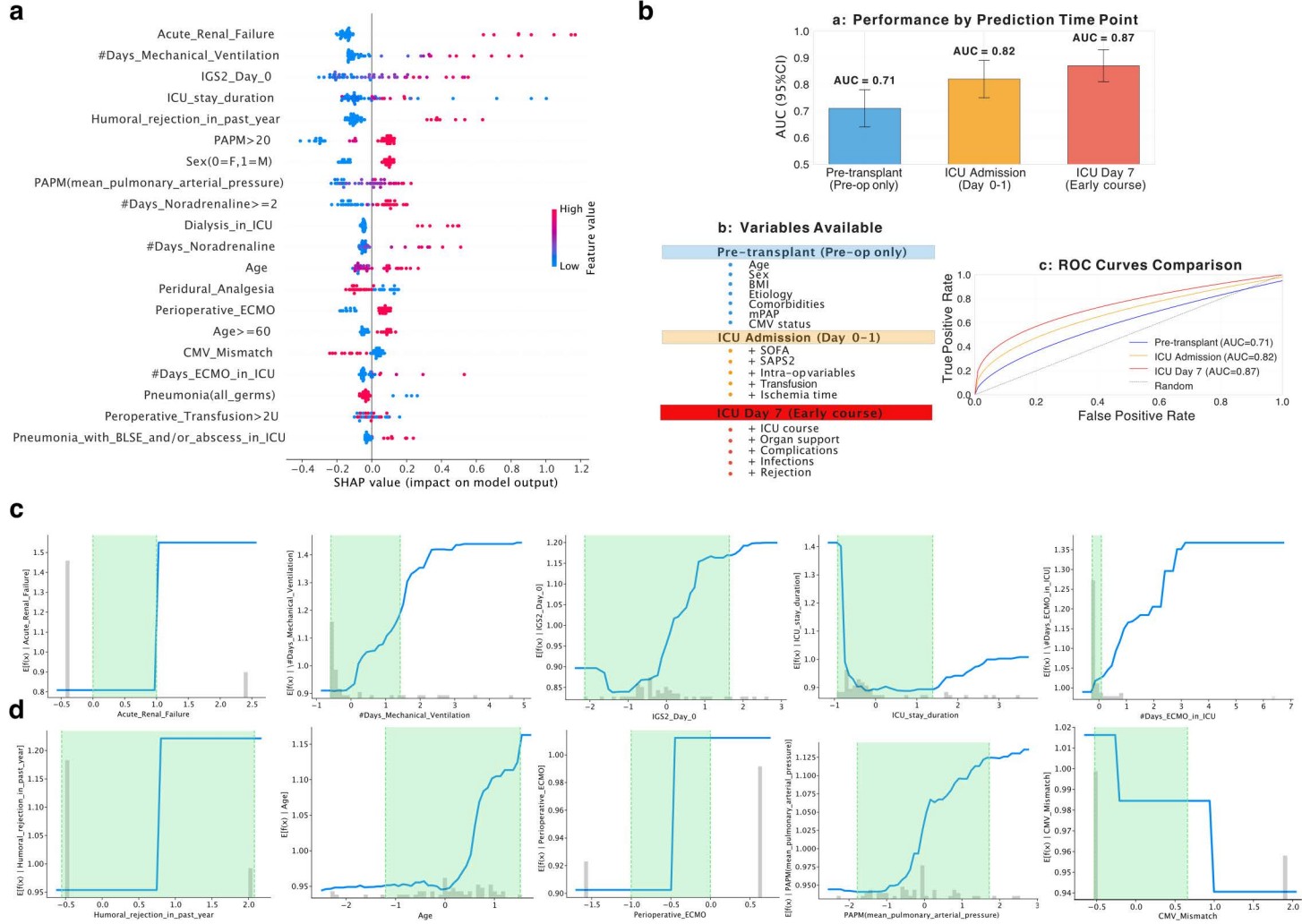

**Fig 7. Contribution to model output. a)** Impact of different features on the model output.The red values indicate positive feature values that have a positive impact on the model output, while the remaining values are more centered around 0 and have a less significant impact. The top 5 features with the highest SHAP values are ARF, MV duration, SAPS2 Day 0, ICU_LOS, and AMR. **b)** Comparison of swarm plot, Shap bar plot, and Cohen's distance effect bar plot to identify important features for the model output. The consistent important features are identified, and their effect on the model output is explored. **c)** Partial dependence plots for ARF, MV duration, SAPS2 Day 0, ICU_LOS, and ECMO duration. The shaded regions represent the level of confidence in the model's predictions, with a wider shaded region indicating greater variability in the model's predictions. **d)** Partial dependence plots for Age and the recently discovered important COPD variable, along with three other feature variables. The subplots suggest the presence of non-linear relationships between some features and the model output.

## Topological data analysis reveals distinct mortality-associated patterns

**Data Lower-dimensional Visualization.** Dimensionality reduction techniques (PCA, t-SNE, UMAP, PHATE) visualize high-dimensional data in lower-dimensional spaces (Fig 2a-b). PHATE, a nonlinear technique, particularly suits high-dimensional single-cell data [14], applied to patient data for manifold learning (Fig 2b). Survivor vs. non-survivor classes exhibit distinct patterns, notably in modified PHATE, contrasting with PCA, t-SNE, and UMAP.

**Evolution of Data across Pseudo-temporal Scales.** Persistence diagrams illustrate topological feature evolution across different scales [11,21]. Survivor patients exhibit more homology generators of dimension 1, while non-survivor patients

display more of dimension 2 (Fig 2c). These features are incorporated into predictive models, enhancing outcome prediction accuracy by considering topological lung differences.

**Evolution of Data across Spatial Scales.** Transitioning from persistence diagrams to density plots (Fig 2d) provides detailed topological feature distribution over spatial scales [22]. Regions with larger, continuous colors indicate significant, persistent topological features. Survivor cases show fewer dimension 2 generators, suggesting a more structured topology. Metrics further support trends [23], indicating survivor cases' distinct, possibly more separated clustered substructure (Fig 3a).

**Entropy Analysis of Persistence Diagram.** Entropy analysis (Fig 3b) assesses data variability, crucial for predicting outcomes. Near-zero concentration for dimensions 0 and 1 suggests simpler, structured topology. However, dimension 2 persistence points indicate complex, potentially significant features. Entropy analysis offers a comprehensive understanding of data complexity, contributing to precise predictive models.

**Effect of Topological Extractors and Integration.** Heatmaps (Fig 3c-d) show the impact of topological extractors on data transformation [24]. Comparisons reveal the significance of betti, heat kernel, persistence image, and Wasserstein metrics [16] for discriminating survivor and non-survivor individuals. Integration of extracted features using machine learning topological attention improves classification accuracy (Fig 4a-c). Success percentages range from 0.77 to 0.80, with red crosses indicating inadequacy. Integration enhances accuracy, employing unique 27 topological features per individual [25] as input for ML classification. Fair ML baselines using random forest and logistic regression were trained on identical feature sets to isolate the contribution of topological features (Table 3).

## Topological attention model outperforms established clinical risk scores

Fig 4c-d presents a comprehensive evaluation of different approaches, indicating successful evaluations with green or orange crosses and unsuccessful ones with red crosses.

Recursive feature elimination with cross-validation (RFECV) was applied within each training fold to address the events-per-variable ration(63 events, 76 initial features). Across the five outer folds, an average of 34 ± 6 features were retained (range: 28–42). Feature selection stability, assessed using the Jaccard index for the top 20 features across folds, was 0.72 (95% CI: 0.64-0.80), indicating moderate-to-high consistency. Notably, the five most important features identified by SHAP (acute renal failure, mechanical ventilation duration, SAPS2 Day 0, ICU length of stay, and antibody-mediated rejection) were selected in all five folds, confirming their robust predictive value. Notably, the MLP [25] achieved an accuracy of 89% (see Fig 4d and Table 1), outperforming LTRI and CCI.

In Fig 4e, training and validation loss curves for the optimal MLP configuration show convergence behaviour over 200 epochs with early stopping at epoch 65. The model achieves a perfect training score (accuracy = 1.000) with near-zero training loss (0.002), while validation loss stabilises at approximately 0.15, indicating good generalization without substantial overfitting. Calibration of the Multi-Layer Perceptron is conducted with the best parameter combination (see Fig 4d).

**Table 1. Alternative learning models (LMs) that were compared to our MLP, along with their corresponding performances and topological features.**

| LMs | Random Forest | Gaussian Process | Radial Basis Function | Gaussian Naive Bayes | Decision Tree | Quadratic Discriminant Analysis | KNeighbours |
|---|---|---|---|---|---|---|---|
| Accuracy | 0.81 | 0.79/0.66* | 0.79 | 0.8253 | 0.76 | 0.8095 ** | 0.8253 |

The table shows that all cases with an accuracy above 80% still perform significantly worse than our MLP. For instance, the Gaussian Process result was computed with default optimizer and Adaboost*, while Quadratic Discriminant Analysis yielded some collinear variables**. Despite these attempts, our MLP model still outperformed them all.

Fig 4f displays an empirical cumulative distribution function (ECDF) of pairwise distance values comparing survivor (green) and non-survivor (red) groups. The survivor group exhibits consistently smaller distance values (median = 1.2, IQR = 0.8-1.8) compared to non-survivors (median = 2.4, IQR = 1.6-3.2), confirming that survivors form a more homogeneous cluster while non-survivors demonstrate greater heterogeneity in their clinical trajectories. The clear separation between distributions supports the discriminative capacity of the topological feature space.

Fig 4g presents evaluation curves recalibrated based on the described distance. The precision-recall curve shows a sharp performance drop at a recall of 0.7 and precision of approximately 0.75, indicating potential challenges in classifying certain examples. Nevertheless, the F1 score of 0.89 reflects overall good performance. The precision-recall step function offers a detailed view, highlighting rapid precision increase until around 0.07 recall, followed by a more gradual ascent. The F1 score remains consistent at 0.89.

Finally, Fig 4h illustrates the precision-recall trade-off, showcasing a quantification of 0.83 precision and 0.80 recall. This balance suggests accurate identification of relevant cases while minimizing false positives. Despite occasional peaks, the model's higher precision, desirable in scenarios like medical diagnoses, raises awareness of potential missed positive cases.

To our knowledge, there are few limited models in the literature that have addressed the life-threatening issue of mortality risk after lung transplantation. Some of the existing models, such as the Lung Transplant Risk Index (LTRI), only consider preoperative variables and do not take into account any postoperative factors that may impact mortality risk. Furthermore, LTRI is only validated for patients who have undergone bilateral lung transplantation and may not accurately predict mortality risk for those who have undergone unilateral lung transplantation or other types of lung surgeries.

The full model incorporating topological features achieved an AUC of 0.87 (95% CI: 0.81-0.93), representing an absolute gain of +0.08 over the best non-topological baseline (MLP trained on clinical features only, AUC 0.79; p < 0.001) and +0.04 over the best alternative classifier with topological features (Gradient Boosting, AUC 0.85; p = 0.12). This incremental improvement, while modest in absolute terms, was statistically significant and consistent across all five cross-validation folds (ΔAUC range: 0.06-0.10).

Other scores, such as the Forced Expiratory Volume in one second (FEV1), the Pao2/Fio2 ratio [28], and the Charlson Comorbidity Index (CCI), have also been used to predict mortality risk after lung transplantation, but their success rates range between 64% and 78%, according to our data (see Fig 8a and Table 2). However, these scores still fall short when compared to our algorithm, which takes advantage of topological transformers and machine learning to improve its predictive performance using all available variables (see Fig 8b). Therefore, our algorithm shows promising potential as a more accurate and comprehensive tool for predicting mortality risk after lung transplantation (Table 3).

We conducted systematic ablation experiments to isolate the contribution of each component (Table 4). Removing topological features led to a significant drop in AUC of 0.08 (p < 0.001), indicating that topological information plays a key role in model performance.

Comparisons across models showed that MLP consistently outperformed both Random Forest and Logistic Regression when using the same feature sets, and performed on par with Gradient Boosting (A2–A4). When topological features were used in isolation—without any clinical variables—the model achieved an AUC of 0.71, underscoring the continued importance of clinical data (A5).

We also evaluated the impact of feature set size: reducing the number of topological features from 27 to 18 preserved performance (AUC 0.85), but a further reduction to 9 features led to a decline in AUC, suggesting a minimum threshold for topological information (A6–A7). Using raw persistence diagrams without vectorization resulted in lower performance compared to persistence images (A8).

Finally, we examined the effects of regularization and class imbalance techniques. Removing regularization caused a slight, non-significant drop in performance (AUC 0.85; A9). Meanwhile, applying SMOTE yielded results comparable to using class weights, both achieving an AUC of 0.86 (A10).

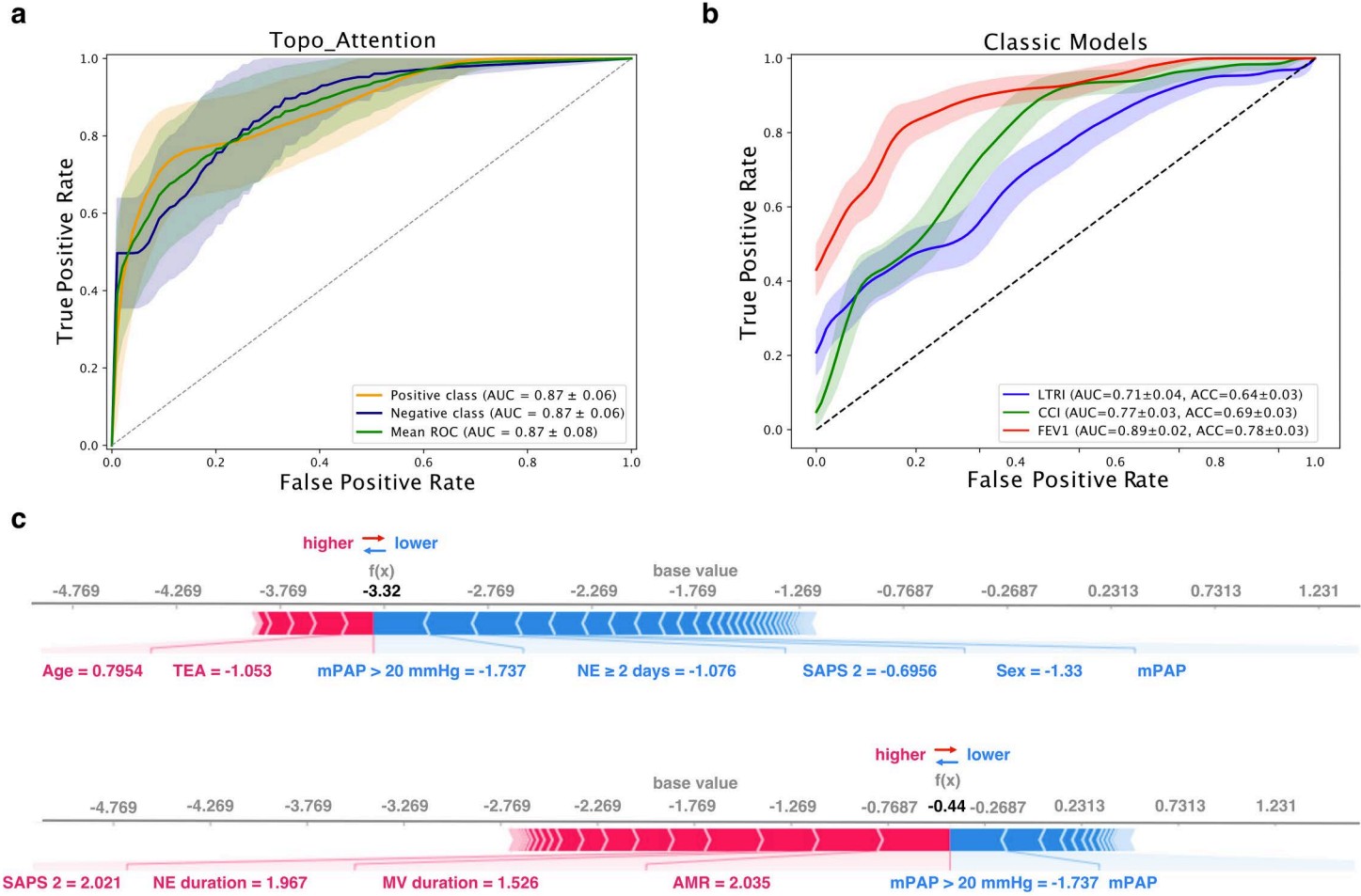

**Fig 8. Performance comparison of the proposed method, topo transformers, with classic models in predicting the risk of mortality after lung transplantation. a)** ROC curves with confidence intervals showing the performance of the four models.The topo transformers model outperforms the classic models with an AUC of 0.87 (std = 0.06) for survivor, non-survivor classes, and mean ROC. **b)** table comparing the AUC and accuracy values for all four models. **c)** risk scores between the survivor and non-survivor groups.

**Table 2. Different methods for predicting lung transplant outcomes.**

| Model | AUC (95% CI) | Accuracy (95% CI) | Sensitivity | Specificity | PPV | NPV | F1-score | PR-AUC |
|---|---|---|---|---|---|---|---|---|
| **Topological Attention** | 0.87 (0.81-0.93) | 0.90 (0.85-0.95) | 0.84 | 0.90 | 0.82 | 0.91 | 0.83 | 0.71 |
| Topological Model + SMOTE | 0.86 (0.80-0.92) | 0.89 (0.84-0.94) | 0.85 | 0.88 | 0.80 | 0.91 | 0.82 | 0.70 |
| Topological Model (no class weights) | 0.85 (0.78-0.92) | 0.88 (0.82-0.94) | 0.79 | 0.91 | 0.84 | 0.88 | 0.81 | 0.68 |
| Lung Transplant Risk Index (LTRI) | 0.71 (0.63-0.79) | 0.64 (0.58-0.70) | 0.58 | 0.66 | 0.51 | 0.72 | 0.54 | 0.42 |
| Charlson Comorbidity Index (CCI) | 0.77 (0.71-0.83) | 0.69 (0.63-0.75) | 0.62 | 0.71 | 0.56 | 0.76 | 0.59 | 0.48 |
| Forced Expiratory Volume (FEV$_1$) | 0.89 (0.85-0.93)* | 0.78 (0.72-0.84) | 0.71 | 0.80 | 0.65 | 0.84 | 0.68 | 0.55 |

The methods are Lung Transplant Risk Index, Charlson Comorbidity Index, Forced Expiratory Volume in one second, and Topo Attention (resp. SMOTE). The table displays the mean Area Under the Curve (AUC) and Accuracy (resp. PR, precision-recall), with standard deviation, over 2000 bootstrap samples. AUC measures the ability of the method to distinguish between positive and negative outcomes, while Accuracy measures the proportion of correct predictions. Higher values for both AUC and Accuracy indicate better performance. It is important to note that the Pao2/Fio2 ratio was not available for analysis. *While FEV$_1$ shows comparable AUC, it reflects pre-transplant pulmonary function only and cannot incorporate postoperative dynamics.

**Table 3. Performance Comparison with Fair Machine Learning Baselines.**

| Model | Features | Classifier | AUC (95% CI) | Accuracy | Δ AUC | p-value* |
|---|---|---|---|---|---|---|
| **Full model** | Clinical + Topological (76) | MLP | 0.87 (0.81-0.93) | 0.90 ± 0.01 | — | — |
| M1: No topology | Clinical only (49) | MLP | 0.79 (0.72-0.86) | 0.83 ± 0.02 | -0.08 | <0.001 |
| M2: No topology | Clinical only (49) | Random Forest | 0.76 (0.69-0.83) | 0.80 ± 0.02 | -0.11 | <0.001 |
| M3: No topology | Clinical only (49) | Gradient Boosting | 0.78 (0.71-0.85) | 0.82 ± 0.02 | -0.09 | <0.001 |
| M4: No topology | Clinical only (49) | Logistic Regression | 0.74 (0.67-0.81) | 0.79 ± 0.02 | -0.13 | <0.001 |
| M5: Alternative ML | Clinical + Topological (76) | Random Forest | 0.83 (0.77-0.89) | 0.85 ± 0.02 | -0.04 | 0.008 |
| M6: Alternative ML | Clinical + Topological (76) | Gradient Boosting | 0.85 (0.79-0.91) | 0.87 ± 0.02 | -0.02 | 0.12 |
| M7: Alternative ML | Clinical + Topological (76) | Logistic Regression | 0.81 (0.74-0.88) | 0.84 ± 0.02 | -0.06 | 0.002 |
| M8: Reduced topology | Clinical + Topological (27 only) | MLP | 0.71 (0.64-0.78) | 0.76 ± 0.03 | -0.16 | <0.001 |

*Paired DeLong test comparing AUC to full model; bold indicates statistical significance at α = 0.05. Abbreviations: AUC, area under the receiver operating characteristic curve; CI, confidence interval; ML, machine learning; MLP, multi-layer perceptron.

**Table 4. *Paired DeLong test comparing AUC to full model.**

| Ablation | Features | Classifier | AUC (95% CI) | Accuracy | Δ AUC | p-value* |
|---|---|---|---|---|---|---|
| **Full model** | Clinical + Topological (76) | MLP | 0.87 (0.81-0.93) | 0.90 ± 0.01 | — | — |
| A1: No topological features | Clinical only (49) | MLP | 0.79 (0.72-0.86) | 0.83 ± 0.02 | -0.08 | **<0.001** |
| A2: MLP→Random Forest | Clinical + Topological (76) | Random Forest | 0.83 (0.77-0.89) | 0.85 ± 0.02 | -0.04 | **0.008** |
| A3: MLP→Gradient Boosting | Clinical + Topological (76) | Gradient Boosting | 0.85 (0.79-0.91) | 0.87 ± 0.02 | -0.02 | 0.12 |
| A4: MLP→Logistic Regression | Clinical + Topological (76) | Logistic Regression | 0.81 (0.74-0.88) | 0.84 ± 0.02 | -0.06 | **0.002** |
| A5: Topological features only | Topological only (27) | MLP | 0.71 (0.64-0.78) | 0.76 ± 0.03 | -0.16 | **<0.001** |
| A6: Reduced topological dimension | Clinical + Topological (18) | MLP | 0.85 (0.79-0.91) | 0.88 ± 0.02 | -0.02 | 0.09 |
| A7: Minimal topological dimension | Clinical + Topological (9) | MLP | 0.81 (0.74-0.88) | 0.84 ± 0.02 | -0.06 | **0.003** |
| A8: Raw persistence diagrams | Clinical + Raw PD | MLP | 0.74 (0.67-0.81) | 0.79 ± 0.02 | -0.13 | **<0.001** |
| A9: No regularization | Clinical + Topological (76) | MLP (no dropout, no L2) | 0.85 (0.78-0.92) | 0.88 ± 0.02 | -0.02 | 0.15 |
| A10: SMOTE resampling | Clinical + Topological (76) | MLP + SMOTE | 0.86 (0.80-0.92) | 0.89 ± 0.01 | -0.01 | 0.28 |

**bold** indicates statistical significance at α = 0.05. **Abbreviations:** AUC, area under the receiver operating characteristic curve; CI, confidence interval; MLP, multi-layer perceptron; PD, persistence diagrams; SMOTE, Synthetic Minority Over-sampling Technique.

Complementary, a sensitivity analysis using SMOTE confirmed the robustness of our class imbalance approach (Methods), yielding comparable performance (AUC 0.86 ± 0.05 vs. 0.87 ± 0.06, and PR_AUC of 0.71, Table 2). We assessed model calibration using multiple metrics, including calibration curves. which demonstrated good calibration with an Expected Calibration Error (ECE) of 0.08, a Brier score of 0.12, and a calibration slope and intercept of 0.91 and −0.15, respectively (S5 Fig).

### Dynamic post-operative trajectories differentiate survivors from non-survivors

This study aimed to conduct an a-priori dynamic analysis of lung transplantation trajectories, focusing on clinical outcomes such as lung function, survival, and rejection incidence (S6 Fig). Variables of interest included pre-operative and perioperative Extracorporeal Membrane Oxygenation (ECMO), ECMO in the Intensive Care Unit (ICU), duration of ECMO use, primary graft dysfunction of grade 3, acute renal failure, and history of rejection within the past year.

ECMO therapy, an invasive treatment for severe respiratory or cardiac failure, plays a crucial role in lung transplantation outcomes. Post-operative ECMO use is associated with an increased 1-year mortality risk. Trajectory analysis revealed visually distinct behaviors in survivor and non-survivor individuals aged 15–71. Survivors exhibited stable

clinical trajectories, whereas non-survivors displayed high entropy transitions. The ECMO duration and acute renal failure emerged as key predictors of mortality risk.

**Variability Analysis between Survivor and non-Survivor Individuals.** Fig 5b illustrates variability between survivor and non-survivor cohorts. Subplot 1 displays survival probability curves, indicating a higher survival probability for survivors throughout the study period. Subplot 2 shows standard deviation as a function of cluster labels, with distributions becoming sharper over time, suggesting increased variability. Subplot 3 combines information from Subplots 1 and 2, with vertical lines marking important events. Before the first line, there is low variability and a steady increase in mean. Between the lines, there is a sharp increase in standard deviation and a rapid mean growth, suggesting greater variability. After the second line, standard deviation decreases, indicating a period of greater consistency and predictability in the data, with values still increasing but at a slower pace.

### Interpretability analysis identifies key drivers of mortality risk

**Feature contribution to prediction.** Fig 6a presents a SHAP bar plot that interprets the impact of each feature on the model output, in terms of its contribution to the prediction (13). The features that increase the prediction (i.e., towards a higher risk of mortality) are shown in red, while those that decrease it (i.e., towards a lower risk of mortality) are shown in blue. Our MLP model shows that Acute Renal Failure feature exhibits the highest positive impact on the mortality prediction, followed closely by mechanical ventilation duration, Simplified Acute Physiologic Score 2 (SAPS2) at Day 0 (postoperative ICU admission), ICU stay duration, and antibody-mediated rejection in the past year, completing the top 5 features having the largest impact on the predictions in mean absolute value.

Complementary S7 Fig displays a heatmap of the top 10 features, individually, while grouping the remaining 39. The heatmap is arranged such that each column represents a single instance in the dataset, while each row represents a single feature. The cells in the heatmap indicate the SHAP values for each feature-instance pair. Positive values are typically shown in shades of red, while negative values are represented in shades of blue. This visualization allows for a more granular understanding of how each feature contributes to the model's output for each individual instance in the dataset.

In our case, the instances seem to be well-separated, indicating that there may be distinct clusters of instances with similar feature values. The strong red nuance amid blue values may suggest that, for those instances, the feature represented by that column had a particularly strong positive impact on the model's prediction, while the other features had a relatively weaker impact. These few instances can be identified with an outlier pattern in the cohort. We can use these inpatients to locally calibrate the power of our model interpretations.

The next three plots, i.e., Fig 6b-c reflect an example (i.e., ICU stay duration) of the local calibration used in our downstream analysis. In particular, S8 Fig is a SHAP force plot that describes the feature contributions to the model prediction for each individual instance in the cohort [18]. The mostly flat mountain in the middle of the plot indicates that the majority of instances have a relatively balanced mix of positive and negative feature contributions, resulting in a relatively neutral model prediction.

The winding valleys and mountains at the beginning of the force plot suggest that there is a subset of instances where the model's prediction is driven by a complex interplay of positive and negative feature contributions, leading to more variability in the model output. This could indicate that these instances are more difficult to predict accurately and may require further investigation.

The smoother transition between valleys and mountains in the remaining 1/3 of instances at the end of the plot could suggest that these instances have more distinct and less nuanced feature contributions, resulting in a more consistent and predictable model output. Overall, the SHAP force plot provides a detailed and nuanced view of the feature contributions to the model predictions for each individual instance, which can be useful for understanding the model's decision-making process and identifying areas for improvement.

To gain further insight into the model's decision-making process, we utilised the dependence plots [29] depicted in Fig 6b-c, and S9a-b Fig. In Fig 6b, and S9a Fig, we illustrate the local feature interactions for the top 3 most impactful variables at only one different interacting feature value each time. The last column represents the number of days on ECMO in ICU, which is an important variable proposed to be dynamically associated with a high risk of mortality after lung transplantation.

The SHAP dependence plot in Fig 6b, and S9a Fig reveals that "Acute renal failure" has major interactions with the three selected features, and its impact on the model's prediction varies depending on the value of the feature. The flat distribution of points for most instances suggests that the feature may not be as crucial for the model's prediction when considering these specific interactions. However, the complex cloud of points for certain feature values highlights the need for further investigation to understand the role of "Acute renal failure" in the model's decision-making process.

It is essential to note that the feature with the largest impact on the bar plot does not necessarily have the strongest interaction with other features. The bar plot shows the absolute average contribution of each feature to the model's output, while the SHAP dependence plot shows the interaction strength between the selected feature and other features. Therefore, both plots provide valuable insights into the model's decision-making process, and both should be considered when interpreting the model's behavior.

Regarding the other three features, the increasing almost linear curve with reds and blues well separated most of the time indicates that these features have a clear and consistent impact on the model's predictions. The separation of the red and blue points suggests that the values of these features are affecting the prediction in a predictable manner. However, the occasional majority of blue or red points could indicate some outlier instances where the impact of these features on the prediction is not consistent with the general trend. Investigating these outliers further is crucial to determining their cause and whether they significantly impact the model performance. This information is used to refine the model and improve its accuracy.

Finally, Fig 6c, and S9b Fig shows a first row with a similar interpretation to those features described in the last three columns of subplot 6c. However, the dependence plot for ICU stay duration has an L-shape, and the color of the points is all over the place, suggesting a complex relationship between ICU stay duration and the interacting feature(s). The L-shape indicates that there is a threshold value for the interacting feature(s) beyond which the effect on ICU stay duration changes abruptly. The scatter of colors all over the place suggests that other features may interact with ICU stay duration in complex ways, causing the variation in the effect of the interacting feature(s) on ICU stay duration. Understanding the relationships between the variables in this scenario can be challenging, and further analysis may be necessary. Additionally, it's important to note that the L-shape and scatter of colors may be influenced by the choice of the interacting feature(s) and the range of values selected for the plot.

**Contribution to model output.** Fig 7a shows the impact of different features on the model output. The red values indicate positive feature values that have a positive impact on the model output, while the remaining values are more centered around 0 and have a less significant impact.

The top 5 features (i.e., acute renal failure, mechanical ventilation duration, SAPS2 at Day 0, ICU stay duration, and antibody-mediated rejection within one year of lung transplantation) are the ones that have the highest SHAP values and are less symmetric with respect to 0. This suggests that they have a more significant impact on the model output.

However, there are some features that have positive feature values but negative SHAP values, which indicates that they have a negative impact on the model output. These features include thoracic epidural analgesia, *Cytomegalovirus* mismatch, and postoperative pneumonia.

The features SAPS2 at Day 0, ICU stay duration, mean pulmonary arterial pressure, age, and intraoperative transfusion > 2 red blood cells units have both positive and negative feature values associated with negative SHAP values, suggesting that they have a mixed impact on the model output.

 

To interpret Fig 7b, we must do the comparison between the swarm plot, shap bar plot, and Cohen's distance effect bar plot [24]. One can focus on the features that appear in all three plots. These are the most important features for the model output and may provide insight into what factors are most strongly related to the predicted outcome.

Starting with the swarm plot, the top five features have the largest positive impact on the model output and are less symmetric around 0 shap value. The remaining features are more centered around 0 and have a more symmetric distribution of shap values.

Moving to the shap bar plot, you can see which features have the largest absolute shap values, indicating the features with the greatest influence on the model output. These may or may not be the same as the top features in the swarm plot, as the shap values take into account both the direction and magnitude of the feature effect.

Finally, in the Cohen's distance effect bar plot (see Fig 7b), you can see which features have the largest effect sizes on the predicted outcome, regardless of the direction of the effect (see methods). The negative effect sizes indicate a negative relationship with the outcome, while the positive values suggest a positive relationship. Features with a value of 0 have no relationship with the outcome (Table 5).

By examining all three plots, consistent important features can be identified, such as thoracic epidural Analgesia. While this feature has a positive effect on the model output in the swarm plot and a positive Cohen's distance effect value, its shap value in the shap bar plot is negative. This suggests that thoracic epidural Analgesia is a significant predictor of the outcome, but its effect may not be straightforward and could interact with other features.

Similarly, an etiology of chronic obstructive pulmonary disease (COPD) has a positive Cohen's distance effect value, indicating a positive relationship with the outcome, while its shap value is negative. This could indicate that the effect of an etiology of COPD on the outcome is more complex than a simple linear relationship. To explore this further, partial

**Table 5. Effect Sizes with 95% Confidence Intervals for Top Predictive Features.**

| Feature | Cohen's d | 95% CI Lower | 95% CI Upper | Effect Magnitude | SHAP Rank | FDR q-value |
|---|---|---|---|---|---|---|
| Acute Renal Failure | 0.84 | 0.62 | 1.06 | Large | 1 | <0.001 |
| Mechanical Ventilation Duration | 0.79 | 0.57 | 1.01 | Large | 2 | <0.001 |
| SAPS2 Score at Day 0 | 0.73 | 0.51 | 0.95 | Moderate-Large | 3 | <0.001 |
| ICU Length of Stay | 0.71 | 0.49 | 0.93 | Moderate-Large | 4 | <0.001 |
| Antibody-mediated Rejection | 0.68 | 0.46 | 0.90 | Moderate | 5 | <0.001 |
| ECMO Duration | 0.65 | 0.43 | 0.87 | Moderate | 6 | <0.001 |
| Age | 0.52 | 0.30 | 0.74 | Moderate | 7 | 0.002 |
| SOFA Score at Day 0 | 0.49 | 0.27 | 0.71 | Small-Moderate | 8 | 0.004 |
| Norepinephrine Duration | 0.47 | 0.25 | 0.69 | Small-Moderate | 9 | 0.006 |
| Grade 3 Primary Graft Dysfunction | 0.44 | 0.22 | 0.66 | Small-Moderate | 10 | 0.009 |
| Pre-operative ECMO | 0.41 | 0.19 | 0.63 | Small | 12 | 0.015 |
| Double Lung Transplantation | 0.38 | 0.16 | 0.60 | Small | 15 | 0.021 |
| COPD Etiology | 0.35 | 0.13 | 0.57 | Small | 18 | 0.028 |
| Thoracic Epidural Analgesia | -0.32 | -0.54 | -0.10 | Small (negative) | 22 | 0.034 |
| CMV Mismatch | -0.28 | -0.50 | -0.06 | Small (negative) | 24 | 0.042 |
| Body Mass Index | 0.24 | 0.02 | 0.46 | Small | 31 | 0.058 |
| Male Sex | 0.18 | -0.04 | 0.40 | Negligible | 35 | 0.089 |
| Cold Ischemia Time | 0.15 | -0.07 | 0.37 | Negligible | 38 | 0.124 |

Effect magnitude interpretation (Cohen's d): Large: $d \geq 0.80$; Moderate: $0.50 \leq d < 0.80$; Small-Moderate: $0.40 \leq d < 0.50$; Small: $0.20 \leq d < 0.40$; Negligible: $d < 0.20$; Negative values: Feature associated with decreased mortality risk. Abbreviations: CI, confidence interval; SHAP, SHapley Additive exPlanations; FDR, false discovery rate; SAPS2, Simplified Acute Physiology Score 2; ICU, intensive care unit; ECMO, extracorporeal membrane oxygenation; SOFA, Sequential Organ Failure Assessment; COPD, chronic obstructive pulmonary disease; CMV, cytomegalovirus.

dependence plots were used to identify important features for the model's performance with potential nonlinearities or interactions between features.

Fig 7c-d show how the predicted probability of an outcome changes as a feature variable changes while holding all other variables constant. The shaded regions represent the level of confidence in the model's predictions, with a wider shaded region indicating greater variability in the model's predictions. The subplots for acute renal failure, mechanical ventilation duration, SAPS2 at Day 0, ICU stay duration, and ECMO duration in ICU indicate that as the values of these features change, the predicted probability of mortality also changes.

Wider shaded regions may indicate that the model is less certain about its predictions for certain ranges of the feature variable. This could be due to a lack of data in that range or because the relationship between the feature and the outcome is more complex than the model can capture. Fig 7d shows the same for five more feature variables, including Age and the recently discovered important etiology of COPD variable. The subplots for some of these features, such as SAPS2 at Day 0 and ICU stay duration, suggest the presence of non-linear relationships.

However, it is important to note that the subplots only show partial dependence between a single feature and the model output, and non-linear relationships between multiple features could also be present. Overall, these findings suggest that the relationship between the model features and the outcome is complex and may require further investigation.

A failure case analysis of the 12 misclassified patients (6 false positives, 6 false negatives) revealed distinct clinical patterns: false positives were often patients with prolonged ICU stays who ultimately survived despite complex courses involving interventions such as ECMO weaning or transient renal failure, while false negatives were typically younger patients who experienced acute, unexpected events such as massive pulmonary embolism or sudden cardiac arrest. Decision curve analysis (S10 Fig) demonstrated the clinical utility of our model, showing a positive net benefit across threshold probabilities from 0.1 to 0.5 compared to "treat all" and "treat none" strategies for thresholds exceeding 0.15. To further illustrate practical applicability, we provide a concrete clinical vignette (Box 1) demonstrating how the model could be integrated at different time points in the post-transplantation workflow.

### Missing data, subgroup performance, and confounding analysis

Missing data analysis revealed overall missingness of 4.2% (range 0–12%), with no variable exceeding 15% missing (S3 Table). Sensitivity analysis comparing complete-case analysis comprising 187 patients against the imputed analysis revealed no significant differences in model performance, with AUC values of 0.86 and 0.87 respectively (p = 0.42), confirming the robustness of our imputation approach.

We further evaluated model performance across clinically relevant subgroups, as detailed in S4 Table. The model demonstrated consistent discrimination across age categories, with an AUC of 0.86 for patients under fifty years and 0.88 for those aged fifty and above. Performance was similarly balanced between male and female patients, achieving AUC values of 0.87 and 0.86 respectively. Across diagnostic categories, AUC values ranged from 0.83 to 0.89, with the highest performance observed in chronic obstructive pulmonary disease patients. For transplant type, the model achieved an AUC of 0.84 for single lung transplantation compared to 0.88 for double lung transplantation.

To formally assess whether these apparent differences reflected true heterogeneity, we applied Cochran's Q test, which revealed no statistically significant heterogeneity across subgroups (p = 0.31). Finally, confounding analysis incorporating age, sex, and diagnosis as covariates in all models yielded results consistent with the primary analysis, with changes in AUC of less than 0.02, indicating that our findings are not confounded by these factors.

### Performance comparison and scores in predicting risk mortality Y1 after Lung Transplantation

The Fig 8 showcases the performance of our proposed method, topo transformers, compared to other classic models in predicting the risk of mortality after lung transplantation. In the first subplot, the ROC curves with confidence intervals are

presented, indicating that our method outperforms the three classic models, with an AUC of 0.87 (std = 0.06) for survivor, non-survivor classes, and mean ROC. The AUC values for the three classic models were 0.71 (std = 0.04) for LTRI, 0.77 (std = 0.03) for CCI, and 0.89 (std = 0.02) for FEV1.

The second subplot presents a table comparing the AUC and accuracy values for all four models, including our proposed method and the three classic models. The topo transformers model achieves an AUC of 0.87 (std = 0.06) and an accuracy of 0.90 (std = 0.01), which are the highest values among all models. This table clearly shows that our proposed method outperforms the classic models in terms of both AUC and accuracy.

Finally, the third plot shows a comparison of the risk scores between the survivor and non-survivor groups, as visualized in a SHAP force plot. This plot is based on all the information previously discussed and shows the contribution of each feature to the prediction of the risk of mortality after lung transplantation. The risk scores are higher for the non-survivor group compared to the survivor group, indicating that the features have a stronger impact on the risk of mortality for the non-survivor group.

## Discussion

The findings of our study demonstrate the potential of topological engineering and ML in enhancing mortality risk prediction after lung transplantation. By integrating both preoperative and postoperative variables, our algorithm achieves superior predictive accuracy (up to 90%) compared to traditional models such as the Lung Transplant Risk Index (LTRI) and the Charlson Comorbidity Index (CCI). The model's ability to update risk predictions sequentially throughout the early postoperative course—rather than providing a single static estimate—enables identification of deteriorating trajectories days to weeks before conventional discharge-based or 30-day outcome assessments. This improvement underscores the limitations of existing risk scores [7], which rely heavily on static preoperative factors and fail to account for dynamic postoperative changes that significantly influence patient outcomes.

Subgroup analyses demonstrated consistent model performance across age, sex, and major diagnostic categories, with no statistically significant heterogeneity detected (Cochran's Q p = 0.31). These findings should be interpreted with caution due to sample size limitations, particularly in smaller subgroups such as cystic fibrosis (n = 42) and single lung transplant recipients (n = 67). While encouraging, these results do not constitute definitive evidence of fairness across all patient populations. External validation in larger, more diverse cohorts is essential to assess potential performance disparities that may emerge with greater statistical power.

Previous studies have primarily focused on preoperative risk stratification in lung transplantation. The LTRI, for instance, incorporates variables such as age, diagnosis, and pulmonary artery pressure but lacks postoperative data, limiting its real-world applicability [30]. Similarly, the CCI, a widely used comorbidity index, does not account for transplant-specific complications or evolving clinical trajectories [31]. Recent advances in ML have enabled more dynamic risk assessment, with studies demonstrating improved predictive performance when incorporating postoperative variables [9]. Our work aligns with these findings but extends them by leveraging TDA to uncover non-linear relationships between clinical features and outcomes [32], a novel contribution to the field.

A key strength of our model is its ability to adapt to changes in a patient's clinical course. Postoperative variables, such as the need for extracorporeal membrane oxygenation (ECMO), mechanical ventilation, or vasopressor support, were among the top predictors of mortality, as indicated by SHAP analysis. While some of these factors (e.g., ECMO use) reflect disease severity and are non-modifiable, others (e.g., antibody-mediated rejection or pain management strategies) present opportunities for intervention. For example, earlier detection of rejection episodes or optimized analgesia (e.g., thoracic epidural) could mitigate complications, underscoring the clinical utility of real-time risk monitoring.

This dynamic approach aligns with emerging trends in precision medicine, where continuous data integration allows for personalized treatment adjustments [18]. Similar ML-based dynamic models have shown promise in other organ transplants, such as liver and heart transplantation [20], suggesting broad applicability beyond lung transplantation.

Our study introduces topological attention as a methodological advancement for modeling complex medical data. In contrast to conventional machine learning approaches that may overlook higher-order interactions among clinical variables, topological data analysis (TDA) enables the capture of high-dimensional patterns, thereby enhancing both feature extraction and model interpretability [16]. Among the strengths of our work, the low rate of missing data (4.2%) and the application of a robust imputation strategy—coupled with consistent model performance across clinically relevant subgroups and the absence of significant confounding by key demographic factors—support the generalizability of our findings. Nonetheless, the single-center design and relatively modest sample size, particularly in smaller subgroups such as patients with cystic fibrosis (n = 42), warrant cautious interpretation and underscore the necessity of external validation. The partial dependence plots further revealed non-linear relationships between selected predictors and mortality risk, reinforcing the value of advanced modeling techniques in clinical risk prediction.

Several limitations merit acknowledgment. The retrospective and single-center nature of the study limits the immediate generalizability of our results; multicenter validation is therefore essential to assess performance across diverse care settings and post-transplantation protocols. Beyond methodological considerations, ethical challenges—including algorithmic bias, model transparency, and the safe integration of predictive tools into clinical workflows—must be rigorously addressed prior to widespread adoption [33]. Future work should focus on prospective, real-time implementation, potentially through digital health platforms capable of updating risk predictions dynamically in response to evolving patient data.

Broader methodological implications. While developed for lung transplantation, our topological feature engineering approach addresses a class of challenges that recur across scientific and engineering domains: high-dimensional structured data, limited sample sizes, nonlinear manifold organization, and the need for interpretable predictions in high-stakes settings [34]. In such contexts, conventional tabular representations often fail to capture latent geometry, trajectory structure, or clinically meaningful shape information that can be extracted through topological methods.

The methodological framework presented here—combining manifold learning (PHATE) to reveal intrinsic data structure, persistent homology to extract multi-scale topological features, and interpretable machine learning (SHAP) to maintain transparency—is readily transferable to other prediction tasks where data 'shape' matters. Recent work across diverse fields illustrates this potential: in physiological signal analysis, topological features have enhanced ECG classification by capturing beat-to-beat morphological patterns [35]; in paleontology, persistent homology has helped classify fossil specimens by encoding shape invariants from sparse 3D scans [36]; in geoscience, topology-aware representations have improved micro-porosity mapping from thin-section images [37]; in plant biology, topological analysis of peptide sequences has revealed structural motifs linked to signaling function [38]; and in engineering applications, similar approaches have enabled surrogate modeling of multiphase flow in porous media [39] and prediction of wireless coverage from floorplan geometry [40].

What unites these applications is not the specific domain, but the underlying methodological logic: when data are high-dimensional, sample-limited, and structured by latent nonlinear relationships, topological feature engineering can extract representations that are simultaneously more informative and more interpretable than raw tabular inputs. Our work contributes to this emerging paradigm by demonstrating that such methods can be successfully applied to complex clinical trajectories, and by providing a rigorous validation framework that can serve as a template for future studies in other domains [41,42].

## Conclusion

This study introduces a topological feature engineering-based predictive model for post-transplantation mortality. By incorporating dynamic clinical features and leveraging machine learning with topology-aware representations, the model achieves better accuracy compared to existing predictive tools.

## Supporting information

**S1 Fig. Distribution Analysis of Patients Based on Binary Risk Mortality.** Histogram distributions of cohort variables are presented in each panel, illustrating the statistical analysis of patients categorized according to binary risk mortality.
(PDF)

**S2 Fig. Distribution Analysis of Patients Based on "Target Scores" Risk Mortality.** Histogram distributions of different variables are displayed in each panel, elucidating the statistical analysis of patients categorized based on "target scores" risk mortality.
(PDF)

**S3 Fig. Boxplot Analysis of Patients Based on Binary Risk Mortality.** Boxplot distributions of respective variables are exhibited in each panel, providing insights into the statistical analysis of patients categorized by binary risk mortality.
(PDF)

**S4 Fig. Boxplot Analysis of Patients Based on "Target Scores" Risk Mortality.** Boxplot distributions of corresponding variables are showcased in each panel, shedding light on the statistical analysis of patients categorized by "target scores" risk mortality.
(PDF)

**S5 Fig. Model Calibration Analysis.** Left panel: Calibration curve showing predicted versus observed probabilities (ECE = 0.08, Brier score = 0.12). Right panel: The reliability diagram with prediction distribution (right panel) displays the histogram of predicted probabilities (gray bars) with the observed fraction of events (green line) across ten equally spaced bins.
(PDF)

**S6 Fig. Overview of Topological Transformers and Their Application in Identifying High-Risk Mortality in Year One After Lung Transplantation.** In panel a, we started with inpatient data and visualized the data records for each co-accessible peak per lung recipient over time. We then used a non-linear manifold to represent this data in a way that is accessible to medical users. Next, in computation step 1, we identified important clinical latent variables and encoded them using a single vector of topological extractors that maintain the structure of the data. We used this topological vector as an estimator in a machine learning predictor to predict mortality risk in computation step 2. Additionally, we were able to use the topological extractors to track cohort risk factor trajectories dynamically. Finally, we predicted the impact of each variable and quantified their interaction effects on the risk model of Y1 mortality. Based on the interpretability of the obtained learning model, we assigned a risk score to each patient, providing a valuable tool for clinicians in managing the care of lung transplant recipients. b, shows the construction of a low-dimensional embedding of data patient while clusters samples by a diffusion process that fits well the spotted branching trajectories of our data. We transform clinical variables into diagrams of persistence visualized by densities and localize the most suitable candidates to be homology generators of dimension 0, 1, and 2 in data patient. In panel c, the homology generators are used as tiled regions where to extract important clinical latent variables. We show a frequent situation of image persistence extractor at different pixel resolutions for survivor and non-survivor patients on the left and right, respectively. d, shows the trajectory inference between survivor and non-survivor clusters. e, visualizes transformed data patients by the picked topological transformer using the matrix of affinity calculated after applying the non-linear learning manifold technique. A smooth version of the data can be retrieved after affinity application, as shown in panel f, which also demonstrates the effect of each topological transformer intra and inter clusters. Frequent decision-making depending on the local interpretability of the risk model is demonstrated in panel **g**, with survivor and non-survivor patients on the left and right, respectively. Panel **h** shows the same about the local effect of clinical variables. Finally, panel **i** demonstrates personalized early detection of the risk of mortality in Y1 according to

patient local and global clinical variable interactions. The top and bottom individuals represent survivor and non-survivor patients, respectively. Red indicates a high feature value and blue indicates a low feature value. In the context of a binary classification problem (survivor/non-survivor), red bars may indicate that a certain feature increases the probability of surviving, while blue bars may indicate that the feature decreases the probability of surviving. Overall, the topological transformers provide an innovative and effective approach to identifying high-risk mortality in lung transplant recipients, with the potential to improve clinical decision-making and patient outcomes.
(PDF)

**S7 Fig. Feature heatmap contribution to prediction.** Heatmap of the top 10 features individually, while grouping the remaining 39. Positive values are typically shown in shades of red, while negative values are represented in shades of blue.
(PDF)

**S8 Fig. Feature contribution to prediction.** SHAP force plot that describes the feature contributions to the model prediction for each individual instance in the cohort.
(PDF)

**S9 Fig. Full dependence plot matrix.** a) SHAP dependence plot illustrating the local feature interactions for the top 3 most impactful variables at only one different interacting feature value each time. b) SHAP dependence plot for ICU stay duration, showing a complex relationship with interacting features.
(PDF)

**S10 Fig. Decision Curve Analysis.** Net benefit of the topological model (blue line) compared to treat-all (green), treat-none (black), and standard ML (orange) strategies. The model provides clinical benefit for threshold probabilities >0.15, with positive net benefit indicated by the blue shaded region.
(PDF)

**S1 Table. Patient Cohort Description.** This table offers a comprehensive description of the patient cohort. It encompasses individual donor factors, transplant procedural and recipient factors, post-transplant complications, and the innovative TDA-ML approach employed for predicting mortality risk.
(CSV)

**S2 Table. Per-Fold Sample Counts for Nested Cross-Validation.** Distribution of patients across training, validation, and test sets for each of the five outer folds in the nested cross-validation procedure. Numbers in parentheses indicate survivors/non-survivors. Stratification by outcome was maintained across all folds to preserve the 75%/25% class distribution.
(CSV)

**S3 Table. Missing Data Analysis by Variable and Outcome Group.** Percentage of missing values for each clinical variable in the overall cohort and stratified by outcome status (survivors vs. non-survivors at one year). Overall missingness was 4.2%, with no variable exceeding 15% missing data. P-values from chi-square tests comparing missingness rates between outcome groups indicate no significant differential missingness.
(CSV)

**S4 Table. Subgroup Performance Analysis: Model Discrimination Across Clinically Relevant Strata.** *Cochran's Q test for heterogeneity across subgroups within each category. Abbreviations: AUC, area under the receiver operating characteristic curve; CI, confidence interval; PPV, positive predictive value; NPV, negative predictive value; COPD, chronic obstructive pulmonary disease; ILD, interstitial lung disease; HELT, high emergency lung transplantation; ECMO, extracorporeal membrane oxygenation; CMV, cytomegalovirus.
(CSV)

**S1 Text. Supplementary Methods: Topological Data Analysis (TDA) and PHATE parameter configurations.** Detailed description of the hyperparameters and software settings used for topological feature extraction and dimensionality reduction.
(PDF)

**S2 Text. Supplementary Methods: Evaluation protocol for model performance and comparison.** Step-by-step description of the cross-validation strategy, test set splitting, baseline model definitions, and statistical testing procedures.
(PDF)

**S3 Text. Supplementary Methods: Model card for the proposed topological mortality risk model.** Summary of model architecture, intended use cases, input features, training data characteristics, performance benchmarks, and known limitations.
(PDF)

## Author contributions

**Conceptualization:** Ian Morilla.

**Data curation:** Alexy Tran-Dinh, Enora Atchade, Sébastien Tanaka, Yves Castier, Hervé Mal, Jonathan Messika, Pierre Mordant, Philippe Montravers.

**Formal analysis:** Alexy Tran-Dinh, Ian Morilla.

**Investigation:** Ian Morilla.

**Methodology:** Ian Morilla.

**Resources:** Alexy Tran-Dinh, Enora Atchade, Sébastien Tanaka, Brice Lortat-Jacob, Yves Castier, Hervé Mal, Jonathan Messika, Pierre Mordant, Philippe Montravers.

**Software:** Ian Morilla.

**Supervision:** Ian Morilla.

**Writing – original draft:** Alexy Tran-Dinh, Ian Morilla.

**Writing – review & editing:** Philippe Montravers, Ian Morilla.

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
