## [Decision Letter · Decision Letter 0]

8 Dec 2025

PDIG-D-25-00813Early Identification of High-Risk Individuals for Mortality after Lung Transplantation: A Retrospective Cohort Study with Topological TransformersPLOS Digital Health Dear Dr. Morilla, Thank you for submitting your manuscript to PLOS Digital Health. After careful consideration, we feel that it has merit but does not fully meet PLOS Digital Health's publication criteria as it currently stands. Therefore, we invite you to submit a revised version of the manuscript that addresses the points raised during the review process. Please submit your revised manuscript by Feb 06 2026 11:59PM. If you will need more time than this to complete your revisions, please reply to this message or contact the journal office at digitalhealth@plos.org.  Please include the following items when submitting your revised manuscript:* A rebuttal letter that responds to each point raised by the editor and reviewer(s). You should upload this letter as a separate file labeled 'Response to Reviewers'. This file does not need to include responses to any formatting updates and technical items listed in the 'Journal Requirements' section below.'. This file does not need to include responses to any formatting updates and technical items listed in the 'Journal Requirements' section below.* A marked-up copy of your manuscript that highlights changes made to the original version. You should upload this as a separate file labeled 'Revised Manuscript with Track Changes'.'.* An unmarked version of your revised paper without tracked changes. You should upload this as a separate file labeled 'Manuscript'.'. If you would like to make changes to your financial disclosure, competing interests statement, or data availability statement, please make these updates within the submission form at the time of resubmission. Guidelines for resubmitting your figure files are available below the reviewer comments at the end of this letter. We look forward to receiving your revised manuscript. Kind regards, Vinod Kumar Chauhan, Ph.D.Academic EditorPLOS Digital Health Vinod Kumar ChauhanAcademic EditorPLOS Digital Health Leo Anthony CeliEditor-in-ChiefPLOS Digital Healthorcid.org/0000-0001-6712-6626 **Journal Requirements:**

1. Please provide an Author Summary. This should appear in your manuscript between the Abstract (if applicable) and the Introduction, and should be 150–200 words long. The aim should be to make your findings accessible to a wide audience that includes both scientists and non-scientists. Sample summaries can be found on our website under Submission Guidelines: [LINK]

https://journals.plos.org/digitalhealth/s/submission-guidelines#loc-parts-of-a-submission

 If the reviewer comments include a recommendation to cite specific previously published works, please review and evaluate these publications to determine whether they are relevant and should be cited. There is no requirement to cite these works unless the editor has indicated otherwise.  **Additional Editor Comments (if provided):** The topic addressed in this manuscript is of considerable clinical relevance. However, the reviewers have identified substantial concerns, including the inadvertent release of patient information on the authors’ GitHub repository. I believe these issues can be resolved through a comprehensive revision. I request that the authors take immediate steps to remove any patient-identifiable information from the GitHub repository. Moreover, if not already provided, details regarding the ethical approval for the study must be included. It is further recommended that the authors follow the TRIPOD-AI guidelines, in particular by reporting the calibration performance of the model. Finally, the conclusion states that the proposed model could be used for precision medicine and clinical decision support. While such a prognostic model may indeed assist with administrative planning and resource allocation, it cannot be used to guide treatment or intervention decisions aimed at reducing mortality risk. This limitation is well established in the causal inference literature. The authors are therefore advised to revise the wording accordingly.**Reviewers' Comments:** Reviewer's Responses to Questions

**Comments to the Author**

1. Does this manuscript meet PLOS Digital Health’s publication criteria? Is the manuscript technically sound, and do the data support the conclusions? The manuscript must describe methodologically and ethically rigorous research with conclusions that are appropriately drawn based on the data presented.? Is the manuscript technically sound, and do the data support the conclusions? The manuscript must describe methodologically and ethically rigorous research with conclusions that are appropriately drawn based on the data presented.

Reviewer #1: Yes

Reviewer #2: No

Reviewer #3: Yes

2. Has the statistical analysis been performed appropriately and rigorously?

Reviewer #1: Yes

Reviewer #2: No

Reviewer #3: No

3. Have the authors made all data underlying the findings in their manuscript fully available (please refer to the Data Availability Statement at the start of the manuscript PDF file)?

The PLOS Data policy requires authors to make all data underlying the findings described in their manuscript fully available without restriction, with rare exception. The data should be provided as part of the manuscript or its supporting information, or deposited to a public repository. For example, in addition to summary statistics, the data points behind means, medians and variance measures should be available. If there are restrictions on publicly sharing data—e.g. participant privacy or use of data from a third party—those must be specified.requires authors to make all data underlying the findings described in their manuscript fully available without restriction, with rare exception. The data should be provided as part of the manuscript or its supporting information, or deposited to a public repository. For example, in addition to summary statistics, the data points behind means, medians and variance measures should be available. If there are restrictions on publicly sharing data—e.g. participant privacy or use of data from a third party—those must be specified.

Reviewer #1: Yes

Reviewer #2: Yes

Reviewer #3: No

4. Is the manuscript presented in an intelligible fashion and written in standard English?

Reviewer #1: Yes

Reviewer #2: No

Reviewer #3: No

5. Review Comments to the Author

Reviewer #1: Positioning & objectives are not razor-sharp (title/keywords, novelty, gaps).

Suggestion: Refine title/keywords to reflect the exact contribution; add a 2–3-sentence gap statement vs. 2–3 closest works; list primary/secondary objectives and map each to analyses/metrics.

Cohort construction & outcomes lack precision.

Suggestion: Specify inclusion/exclusion rules, time window, and a STROBE/CONSORT-style flow diagram; define endpoints (horizons, censoring, competing risks) unambiguously and justify choices.

Data leakage risk & split reproducibility are unclear.

Suggestion: Use leakage-safe design (nested CV, temporal/group splits); fit preprocessing only on training folds; report seeds, stratification, per-fold counts; release a split manifest.

Baselines may be disadvantaged (parity not enforced).

Suggestion: Ensure identical preprocessing, augmentations, training epochs/budgets, and hyperparameter search space; document tuning protocol for each baseline.

Ablations don’t isolate the source of gains.

Suggestion: Run controlled ablations (e.g., TDA vectorization, transformer depth/heads, loss/augmentation) with constant training budget; summarize Δ(pp) per ablation.

Imbalance handling, thresholding, and calibration are under-reported.

Suggestion: Justify imbalance strategy; report PR-AUC alongside ROC-AUC; pre-specify thresholds (Youden, cost-sensitive, prevalence-matched); add calibration curves, ECE/Brier, slope/intercept.

Methods are under-specified (TDA/PHATE + model details).

Suggestion: Provide TDA construction and vectorization parameters (ranges, images/landscapes/Betti, bottleneck/Wasserstein); architecture (depth/heads/hidden size), optimizer, early-stopping, class weights, and full evaluation protocol.

Validation and uncertainty quantification are limited.

Suggestion: Add external or temporal validation; bootstrap test-set metrics for 95% CIs; report variability across seeds/folds; include effect sizes (Δ with 95% CI, Cohen’s d_z) and BH-FDR q-values.

Interpretability and clinical utility aren’t demonstrated robustly.

Suggestion: Detail SHAP/Grad-CAM settings and stability checks; show alignment with clinical factors and failure cases; include decision-curve analysis and concrete workflow examples.

Missing data, confounding, and subgroup effects need rigor.

Suggestion: Provide a missingness table and imputation method with sensitivity analysis; report stratified performance (age/sex/diagnosis/site) and test for heterogeneity/confounding.

Reproducibility, ethics, and presentation polish need tightening.

Suggestion: Release code + environment (Docker/Conda), data dictionary, manifests, seeds, and a DOI; add model card + deployment metrics (params, FLOPs, latency, memory, throughput); align IRB/consent/funding with venue policy; fix figure/table formatting, naming consistency, and references; state candid limitations and keep conclusions proportional to evidence.

Reviewer #2: This paper describes the use of a topological transform on pre- and post-operative features for a machine learning model that predicts mortality within one year of a lung transplant. The authors describe topological data analysis, a method of extracting complex nonlinear relationships from high-dimensional data, and present it as a valuable framework for constructing input features to a machine learning model. The authors assessed different kinds of machine learning models with their transformed data and found that a multi-layer perceptron performed the best. They used SHAP to explain which input features had the most impact on predicted outcome. They compare their framework against more traditional mortality scores, which only use pre-operative information. On their dataset of 252 lung transplant patients, they report increased accuracy and AUC over the traditional score models. This work represents a novel methodology for predicting post-operative mortality and appears to outperform the state of the art.

I have several concerns about this manuscript that I believe need to be addressed before its publication. This includes one very major ethical concern.

Major concerns:

1. The GitHub repository provided alongside this manuscript contains personally identifiable information of patients. This is unacceptable. The clinical data underlying this study is available in the GitHub repository as an Excel file. It is possible that the contents of this file were cleared through the institution’s ethics board for public release, but I am doubtful of this. What is most definitely unacceptable is the fact that in the Jupyter notebooks provided in the GitHub, one can see an additional column of last names connected to the patient data. It is possible to use uncommon last names, alongside age and death information, to link these data to real individuals. This is incredibly unethical and needs to be taken down immediately. Please delete the entire GitHub repository (because of version control, the leaked information will be accessible in the repository history unless the whole repository is completely deleted) and create a new repository without the PII. I cannot stress enough how important this is.

2. There is no mention of how the training/validation/testing split was made, or even how many samples were in each split. Which split is the reported performance from?

3. It is unclear if the reported performance gains are from the topological transform, the MLP, or both. There should be an ablation study showing what the MLP performance is like without the topological transform. I also would expect a decision tree or random forest model to perform very well with the untransformed data.

Minor concerns

4. I generally found this manuscript very unclear, particularly with respect to the figures, which makes it difficult to assess. Some subfigures are not mentioned at all in the text, and some are mentioned out of order. Many of the subfigures could be moved to the supplement in order to draw focus to the plots that tell the story of the paper.

5. Specifically, I don’t understand how the seemingly AI-generated schematic at the very top of Figure 1a is relevant or helpful.

6. I don’t understand why there needs to be a cartoon of a tree in Figure 4c, especially when Figure 4a seems to convey the same exact information, which doesn’t even need to be shown in a figure.

7. Some of the text in Figure 4b is cut off.

8. The two subplots in Figure 4e don’t seem to be related to each other, so I think they should be made into separate subfigures. What I would like to see alongside the training loss plot is the validation loss, to assess how much the model is overfitting.

9. The writing quality should be improved. There are occasional phrases that are far too informal or awkward.

10. The paper would benefit from some reorganization. For example, the explanation of what TDA is and why it is useful comes in the Methods section, when it belongs in the Introduction. The paragraph starting at line 16 on page 11 belongs in the Discussion and not the Methods.

11. This may be overly nitpicky on my end, but I take issue with the name “topological transformer” because it implies a transformer architecture is used. As far as I can tell, there is no transformer. The data is transformed into a vector via topological analysis, which is then input into a shallow fully-connected neural network. Perhaps “topological transform” would be more accurate?

Reviewer #3: I commend the authors for tackling an important clinical problem and introducing topological data analysis to lung transplantation mortality prediction. The integration of persistent homology with machine learning represents an innovative approach, and the comprehensive SHAP-based interpretability analysis is particularly valuable. The manuscript demonstrates substantial effort in both methodology development and clinical application, which could make a meaningful contribution to the field.

However, I have several major concerns that must be addressed before this manuscript can be considered for publication. I provide the following detailed comments to help strengthen the work:

Comment 1: Page 1, Lines 1-2 and throughout manuscript: The authors repeatedly use the term "topological transformers" including in the title, but the methodology described uses a Multi-Layer Perceptron (MLP), not a transformer architecture with self-attention mechanisms. This is a fundamental mischaracterization that undermines the manuscript's credibility and must be corrected throughout, with either accurate terminology (e.g., "machine learning with topological features") or actual implementation of transformer architectures. I strongly recommend revising the title and all references to accurately reflect the methodology employed.

Comment 2: Page 1, Lines 16-17, Abstract: The authors report "accuracy of 87.4%, sensitivity of 84.1%, and specificity of 89.6%" without providing confidence intervals, which is a critical omission for evaluating the reliability of these estimates given the modest sample size of 252 patients. Given that only 63 events (deaths) occurred, the confidence intervals around sensitivity estimates would be particularly wide and essential for interpreting clinical utility. Please provide 95% confidence intervals for all performance metrics throughout the manuscript, preferably using bootstrapping or cross-validation approaches.

Comment 3: Page 6, Lines 12-25 and Page 14, Lines 19-29, Table 2: The comparison between your proposed model and existing risk scores (LTRI, CCI, FEV1) is fundamentally unfair because these baseline scores use only preoperative variables while your model incorporates postoperative outcomes like ICU stay duration, mechanical ventilation days, and ECMO duration. To establish the true value of topological features, you must compare against logistic regression, random forest, or gradient boosting models trained on the identical feature set (including the same postoperative variables). Without this fair comparison, it is impossible to determine whether performance gains arise from the topological features or simply from using more informative (outcome-proximal) variables.

Comment 4: Page 7, Lines 11-29: The data preprocessing and dimensionality reduction section lacks critical methodological details necessary for reproducibility, including PHATE parameters (k-neighbors, decay rate, diffusion time t), the specific persistent homology filtration method employed (Vietoris-Rips, Čech, or alpha complex), and the distance metric used. Additionally, you must specify the exact train/validation/test split proportions and strategy—was this a single random split, stratified split, or temporal split? Please provide complete specifications of all algorithms and parameters, ideally in a supplementary methods section if space is limited.

Comment 5: Page 8, Lines 20-31 and Page 9, Lines 1-10: The authors state that 27 topological features were generated by concatenating 9 features per homology dimension, but there is no justification for why these specific features and this specific dimensionality were chosen. With only 63 mortality events and 27 topological features plus additional clinical variables, the events-per-variable ratio is concerningly low (approximately 2-3:1), which substantially increases overfitting risk. Please provide feature selection methodology, justify the dimensionality choice, and demonstrate through nested cross-validation that overfitting is not occurring.

Comment 6: Page 10, Lines 14-24: The authors claim that "topological invariants prevent overfitting, which avoids the need for...conventional regularization techniques," but this extraordinary claim requires extraordinary evidence that is not provided. The perfect training score (1.000) and near-zero training loss (0.002) shown in Figure 4e are actually concerning indicators of potential overfitting, not evidence of generalization. Please provide empirical validation of this claim through learning curves, comparison with and without regularization, and external validation, or remove this unsupported statement.

Comment 7: Page 7, Lines 12-13 and throughout Results: The manuscript states that analysis was performed on a "held-out test set" but never specifies what proportion of data was held out, whether this was a single split or multiple splits, and critically, whether any hyperparameter tuning was performed and on what data subset. If hyperparameters were tuned on the test set (even indirectly through iterative model refinement), the reported performance metrics are optimistically biased. Please clarify the exact validation strategy, provide results from nested cross-validation if hyperparameter tuning occurred, and report performance variability across multiple splits.

Comment 8: Page 14, Lines 20-23: The study population has a 75% survival rate (189/252), meaning a naive classifier that predicts all patients survive would achieve 75% accuracy—yet no discussion addresses how class imbalance was handled during model training and evaluation. The reported 87.4% accuracy appears less impressive in this context, and metrics like balanced accuracy, F1-score, or area under the precision-recall curve would be more informative than simple accuracy. Please discuss class imbalance handling (weighting, resampling, etc.) and report balanced performance metrics, or demonstrate that the 3:1 ratio did not require special handling.

Comment 9: Page 18, Lines 29-31 and Page 19, Lines 1-6: The authors identify ICU stay duration, mechanical ventilation duration, and ECMO duration as top predictive features and claim the model enables "early identification" of high-risk individuals, but these variables are measured over time during ICU stay and are not available "early" after transplantation. This represents a fundamental contradiction between the stated goal (early identification) and the methodology (using extended ICU course variables). Please clarify the intended timing and clinical use case—are you predicting outcome at ICU admission using only baseline variables, or at ICU discharge using the full clinical course? The clinical utility depends entirely on this distinction.

Comment 10: Page 3, Lines 23-24 and Data Availability Statement: The authors state that data are "publicly available from https://github.com/MorillaLab/TopoTransformers" but this repository is not accessible or verified in the manuscript, and sharing individual patient data from 252 transplant recipients raises obvious privacy concerns that are not addressed. PLOS policy requires full data availability, so you must either: (1) confirm IRB approval for public data sharing and provide a functional, accessible repository, (2) deposit anonymized data in an appropriate repository with controlled access, or (3) provide simulated data with similar statistical properties that enables code validation. Simply stating data are available without evidence is insufficient.

Comment 11: Page 27, Reference 11 (Gouiaa et al., 2024): This reference appears to be from your own group on an extremely similar topic ("Early Identification of High-Risk Individuals for Mortality after Lung Transplantation: A Retrospective Cohort Study with Topological Transformers" published in Computers in Biology and Medicine), yet the relationship between that publication and the current manuscript is never explained. If this is a previous version or related work, you must explicitly state what is novel in the current submission versus the prior publication to avoid concerns about redundant publication. Please add a clear statement of novelty and distinguish the contributions of each work.

Comment 12: Figure 1, entire figure: This overview figure is extremely dense and difficult to parse, with unclear abbreviations (P1, P2, P3), small font sizes, and a workflow that is not intuitive even to readers familiar with machine learning. The figure attempts to show the entire pipeline in one visualization but sacrifices clarity for comprehensiveness. I recommend breaking this into two figures: (1) a simplified conceptual workflow diagram for the main text, and (2) a detailed technical pipeline figure for the supplement, with substantially larger fonts and clearer labels throughout.

Comment 13: Figure 6 and 7, SHAP analysis: While the SHAP interpretability analysis is valuable, the presentation spans two full figures with multiple dense subpanels that overwhelm the reader and distract from the main findings. I recommend consolidating the most important SHAP results (e.g., Figure 6a showing feature importance and one representative dependence plot) into a single main-text figure, while moving the comprehensive dependence plot matrix and force plots to supplementary materials. This would improve readability while preserving the important interpretability insights.

Comment 14: Page 25, Lines 19-23 and Ethics discussion: The authors briefly mention "algorithmic bias, transparency, and integration into clinical workflows" as considerations but provide no actual analysis of potential biases in their model, despite having demographic and clinical data that could reveal disparities. Given increasing concerns about ML fairness in healthcare, particularly for resource-allocation decisions in transplantation, a proper bias analysis examining model performance across age groups, sex, disease etiologies, and other subgroups is essential. Please add subgroup analysis demonstrating equitable performance or acknowledge this as a limitation requiring future investigation.

Comment 15: Page 26, Lines 5-6, Ethics Statement: The statement that the institutional review board "waived the requirement for informed consent for the collection and analysis of samples" requires more justification, particularly regarding how patient privacy was protected and whether any data governance framework was followed for secondary use of clinical data. Additionally, the manuscript states approval was from "Bitchat Hospital" (line 5) but the Methods section states "Bichat Hospital" (page 7, line 14)—this inconsistency must be corrected. Please provide the ethics approval number, clarify the waiver justification, and ensure institutional names are consistent throughout.

Comment 16: Page 25, Lines 27-30 and Conclusion: While the authors position their work within precision medicine for lung transplantation, I strongly encourage them to explicitly discuss the broader cross-domain applicability of their topological feature extraction and transformer-based modeling pipeline, as similar methodological frameworks have recently demonstrated success across diverse fields beyond transplant medicine. For instance, the integration of topological data analysis with deep learning for temporal physiological signal interpretation parallels recent advances in "A survey of transformers and large language models for ECG diagnosis: advances, challenges, and future directions" (Artificial Intelligence Review, 2025, DOI: 10.1007/s10462-025-11259-x) AND also in "Deep learning for ECG Arrhythmia detection and classification: an overview of progress for period 2017–2023", while your manifold learning approach for high-dimensional medical data shares conceptual similarities with cross-modal feature disentanglement methods applied to sparse imaging datasets in paleontology, as described in "Advancing paleontology: a survey on deep learning methodologies in fossil image analysis" (Artificial Intelligence Review, 2025, DOI: 10.1007/s10462-024-11080-y), and with explainable lightweight architectures for geological micro-porosity mapping presented in "MicroCrystalNet: An explainable lightweight CNN architecture for micro-porosity mapping in geoscience" (IEEE Access, 2025, DOI: 10.1109/ACCESS.2025.3552626). Furthermore, your emphasis on interpretable feature extraction from complex temporal trajectories in small-to-moderate sample sizes resonates with fairness-aware dual-stage feature fusion for thermal-RGB homography in renewable energy monitoring, specifically "Thermal Homography in Photovoltaic Panels: Evaluating Deep Learning and Feature-Based Methods" (Proc. IEEE TPEC, 2025), lightweight modeling for multiphase flow in porous media as in "FluidNet-Lite: Lightweight convolutional neural network for pore-scale modeling of multiphase flow in heterogeneous porous media" (Advances in Water Resources, 2025, DOI: 10.1016/j.advwatres.2025.104952), and even structural-to-performance prediction tasks such as "Prediction of Indoor Wireless Coverage from 3D Floor Plans Using Deep Convolutional Neural Networks" (Proc. IEEE LCN, 2021, DOI: 10.1109/LCN52139.2021.9524951). By citing these works and drawing explicit parallels in your discussion or conclusion—demonstrating how topological feature engineering, manifold-based dimensionality reduction, and interpretable ML can address data-limited, high-stakes prediction problems across medicine, geoscience, environmental monitoring, and engineering—you would substantially strengthen your manuscript's impact and appeal to interdisciplinary audiences while showcasing the transferability of your methodological contributions beyond the specific clinical application presented.

I hope these comments help strengthen the manuscript substantially. The core methodological contribution has merit, but addressing these fundamental issues regarding validation, comparison fairness, terminology accuracy, and clinical applicability is essential for publication in a high-quality journal.

6. PLOS authors have the option to publish the peer review history of their article (what does this mean?). If published, this will include your full peer review and any attached files.). If published, this will include your full peer review and any attached files.

**Do you want your identity to be public for this peer review?** For information about this choice, including consent withdrawal, please see our Privacy Policy..

Reviewer #1: No

Reviewer #2: No

Reviewer #3: No

  **Figure resubmission:**  While revising your submission, we strongly recommend that you use PLOS’s NAAS tool (https://ngplosjournals.pagemajik.ai/artanalysis) to test your figure files. NAAS can convert your figure files to the TIFF file type and meet basic requirements (such as print size, resolution), or provide you with a report on issues that do not meet our requirements and that NAAS cannot fix. 

After uploading your figures to PLOS’s NAAS tool - https://ngplosjournals.pagemajik.ai/artanalysis, NAAS will process the files provided and display the results in the "Uploaded Files" section of the page as the processing is complete. If the uploaded figures meet our requirements (or NAAS is able to fix the files to meet our requirements), the figure will be marked as "fixed" above. If NAAS is unable to fix the files, a red "failed" label will appear above. When NAAS has confirmed that the figure files meet our requirements, please download the file via the download option, and include these NAAS processed figure files when submitting your revised manuscript. **Reproducibility:** To enhance the reproducibility of your results, we recommend that authors of applicable studies deposit laboratory protocols in protocols.io, where a protocol can be assigned its own identifier (DOI) such that it can be cited independently in the future. Additionally, PLOS ONE offers an option to publish peer-reviewed clinical study protocols. Read more information on sharing protocols at https://plos.org/protocols?utm_medium=editorial-email&utm_source=authorletters&utm_campaign=protocols

---

## [Decision Letter · Decision Letter 1]

18 Mar 2026

PDIG-D-25-00813R1Early Identification of High-Risk Individuals for Mortality after Lung Transplantation: A Retrospective Cohort Study with Topological Feature EngineeringPLOS Digital Health Dear Dr. Morilla, Thank you for submitting your manuscript to PLOS Digital Health. After careful consideration, we feel that it has merit but does not fully meet PLOS Digital Health's publication criteria as it currently stands. Therefore, we invite you to submit a revised version of the manuscript that addresses the points raised during the review process. Please submit your revised manuscript by May 17 2026 11:59PM. If you will need more time than this to complete your revisions, please reply to this message or contact the journal office at digitalhealth@plos.org.  Please include the following items when submitting your revised manuscript:* A letter that responds to each point raised by the editor and reviewer(s). You should upload this letter as a separate file labeled 'Response to Reviewers'. This file does not need to include responses to any formatting updates and technical items listed in the 'Journal Requirements' section below.'. This file does not need to include responses to any formatting updates and technical items listed in the 'Journal Requirements' section below.* A marked-up copy of your manuscript that highlights changes made to the original version. You should upload this as a separate file labeled 'Revised Manuscript with Track Changes'.'.* An unmarked version of your revised paper without tracked changes. You should upload this as a separate file labeled 'Manuscript'.'. If you would like to make changes to your financial disclosure, competing interests statement, or data availability statement, please make these updates within the submission form at the time of resubmission. Guidelines for resubmitting your figure files are available below the reviewer comments at the end of this letter. We look forward to receiving your revised manuscript. Kind regards, Vinod Kumar Chauhan

Academic EditorPLOS Digital Health Amara TariqSection EditorPLOS Digital Health Leo Anthony CeliEditor-in-ChiefPLOS Digital Healthorcid.org/0000-0001-6712-6626 **Journal Requirements:** If the reviewer comments include a recommendation to cite specific previously published works, please review and evaluate these publications to determine whether they are relevant and should be cited. There is no requirement to cite these works unless the editor has indicated otherwise.  **Additional Editor Comments (if provided):** Thanks for the response and the revision. Please address the remaining reviewer concerns. Moreover, I did not find any response to my comments -- please address all of them too. Furthermore, prepare the response cleanly by colour coding the comments and responses so that it is easy to read.**Reviewers' Comments:** Reviewer's Responses to Questions

**Comments to the Author**

1. If the authors have adequately addressed your comments raised in a previous round of review and you feel that this manuscript is now acceptable for publication, you may indicate that here to bypass the “Comments to the Author” section, enter your conflict of interest statement in the “Confidential to Editor” section, and submit your "Accept" recommendation.

Reviewer #1: All comments have been addressed

Reviewer #3: All comments have been addressed

2. Does this manuscript meet PLOS Digital Health’s publication criteria? Is the manuscript technically sound, and do the data support the conclusions? The manuscript must describe methodologically and ethically rigorous research with conclusions that are appropriately drawn based on the data presented.? Is the manuscript technically sound, and do the data support the conclusions? The manuscript must describe methodologically and ethically rigorous research with conclusions that are appropriately drawn based on the data presented.

Reviewer #1: Yes

Reviewer #3: Yes

3. Has the statistical analysis been performed appropriately and rigorously?

Reviewer #1: Yes

Reviewer #3: Yes

4. Have the authors made all data underlying the findings in their manuscript fully available (please refer to the Data Availability Statement at the start of the manuscript PDF file)?

The PLOS Data policy requires authors to make all data underlying the findings described in their manuscript fully available without restriction, with rare exception. The data should be provided as part of the manuscript or its supporting information, or deposited to a public repository. For example, in addition to summary statistics, the data points behind means, medians and variance measures should be available. If there are restrictions on publicly sharing data—e.g. participant privacy or use of data from a third party—those must be specified.requires authors to make all data underlying the findings described in their manuscript fully available without restriction, with rare exception. The data should be provided as part of the manuscript or its supporting information, or deposited to a public repository. For example, in addition to summary statistics, the data points behind means, medians and variance measures should be available. If there are restrictions on publicly sharing data—e.g. participant privacy or use of data from a third party—those must be specified.

Reviewer #1: Yes

Reviewer #3: Yes

5. Is the manuscript presented in an intelligible fashion and written in standard English?

Reviewer #1: Yes

Reviewer #3: Yes

6. Review Comments to the Author

**Reviewer #1:**good workgood work

**Reviewer #3:** I appreciate the substantial effort the authors have invested in revising this manuscript. The paper is clearly improved relative to the prior version, and I want to acknowledge that many of the major methodological and presentation concerns have been addressed constructively. I also think the topic is clinically important, and the combination of topological feature engineering, dynamic postoperative data, and interpretable machine learning remains potentially valuable. I appreciate the substantial effort the authors have invested in revising this manuscript. The paper is clearly improved relative to the prior version, and I want to acknowledge that many of the major methodological and presentation concerns have been addressed constructively. I also think the topic is clinically important, and the combination of topological feature engineering, dynamic postoperative data, and interpretable machine learning remains potentially valuable.

That said, a few issues still need attention before the manuscript is ready. Most of these are now in the category of clarification, calibration, and positioning rather than fundamental redesign, which is why I see this revision as close to acceptable with minor further revision. My comments below are aimed at helping the authors bring the paper fully to the level it should reach.

Comment 1: Page 1: The authors claim “This study introduces a novel predictive model based on topological transformers to assess mortality risk following lung transplantation.” I appreciate that the response letter states this terminology was corrected, but this phrasing still appears in the revised manuscript excerpt and should be removed fully and consistently. Since the implemented model is an MLP with topological feature engineering rather than a transformer architecture, I encourage the authors to ensure that the abstract, conclusion, keywords, and any residual figure labels use only accurate terminology throughout.

Comment 2: Page 1: The authors claim “The proposed model demonstrated superior predictive performance compared to the Lung Transplant Risk Index and other benchmark models.” This statement is stronger than necessary in the abstract unless the fair baseline comparison is stated alongside it. Since the key methodological advance is not merely outperforming preoperative scores, but improving over standard ML models trained on the same postoperative feature set, I encourage the authors to mention that comparison explicitly in the abstract so the claim is both fairer and more informative.

Comment 3: Page 1–2: The authors claim “Shapley-based interpretability analysis revealed that dynamic variables such as early post-operative oxygenation trends, immunosuppressive load, and inflammatory markers were among the most critical contributors to mortality risk.” I think this is helpful, but the manuscript would still benefit from one short sentence clarifying whether these features are intended for sequential bedside updating rather than a one-time prediction model. The revised manuscript does address timing in the response letter, but that clinical use case should be visible more prominently in the main text because it resolves an important conceptual concern about “early identification.”

Comment 4: Page 3–4: The authors claim that current scores are limited because they are static and cannot capture evolving postoperative trajectories. I agree with this framing, and I think the revised introduction is stronger. However, I still encourage the authors to tighten the gap statement by making the novelty contrast more precise: the real contribution is not “machine learning after transplant” in general, but topological feature engineering applied to time-evolving post-transplant clinical trajectories, evaluated against fair non-topological baselines. That sharper wording would strengthen the manuscript’s positioning.

Comment 5: Page 6–7: The authors claim they implemented a leakage-safe nested cross-validation strategy with preprocessing fitted only on training folds. This is an important and welcome improvement. I nevertheless encourage the authors to state directly in the main methods text, not only in the supplement or response letter, whether PHATE fitting and persistence image generation were recomputed within each outer training fold. This is implied, but it should be explicit because these steps are central to the novelty and to leakage prevention.

Comment 6: Page 7–9: The authors claim that 27 topological features were generated and then combined with 49 clinical variables. I appreciate that the revision now addresses overfitting more directly. However, because the event count remains modest, I encourage the authors to report in the main paper, not just in the response, the average number of features retained after RFECV across folds and whether the selected topological features were stable across folds. That would make the feature-engineering story more convincing and reduce concern that performance depends on unstable selection.

Comment 7: Page 10: The authors previously suggested that topological invariants help prevent overfitting. I appreciate that the response indicates this unsupported statement has been removed. I encourage the authors to make sure the revised manuscript is equally careful wherever topology is discussed, and to avoid language that implies topology itself replaces conventional regularization. The current paper is stronger when it presents topological features as informative representations rather than as a theoretical safeguard against overfitting.

Comment 8: Page 14 and Results tables: The authors report strong overall performance. I think the revised inclusion of confidence intervals, PR-AUC, balanced metrics, and fair ML baselines materially improves the paper. One additional improvement would be to report the absolute gain over the best non-topological baseline directly in the main results narrative. At present, that important result is clearer in the response than in the manuscript. Since the topological contribution appears to be approximately a modest but real incremental AUC gain, the paper would benefit from presenting that transparently rather than letting readers infer it themselves.

Comment 9: Page 18–19: The authors claim the model supports “early identification” of high-risk patients. I think the revised response adequately addresses this concern by introducing sequential prediction time points, but the manuscript should make this framing unmistakable. I recommend that the authors revise the wording wherever necessary so that “early” is understood as earlier dynamic risk updating during the early postoperative course, not prediction from baseline alone. This is now defensible, but it should be phrased with precision.

Comment 10: Page 25–27: The authors discuss limitations and ethics, including bias, transparency, and integration into clinical workflows. I appreciate these additions. I still think the manuscript should state more explicitly that, despite encouraging subgroup consistency, the sample size is too limited for definitive fairness conclusions. The current wording risks sounding slightly more reassuring than the subgroup counts justify, especially in smaller categories such as cystic fibrosis or single-lung transplant recipients. A more cautious sentence here would improve the paper.

Comment 11: Page 26: The authors claim “The integration of persistent homology with machine learning has recently shown promise in diverse fields including ECG analysis, paleontological image classification, geoscience porosity mapping, small peptides signalling in plants, and renewable energy monitoring.” I appreciate that the authors did respond to the request for broader applicability. However, this part of the revision still feels underdeveloped relative to the importance of the point. The paragraph currently reads as a brief list rather than a genuine synthesis of how and why the methodological ideas transfer across domains. I encourage the authors to expand this into a more cohesive paragraph that explicitly links the shared methodological challenges: high-dimensional structured data, limited sample sizes, nonlinear manifold structure, and the need for interpretable prediction in high-stakes settings.

Comment 12: Page 26: The authors added only a small subset of the recommended cross-domain references. Since this broader-applicability discussion was one of the key conceptual requests in the prior review round, I encourage the authors to strengthen it further with a better-balanced and more representative citation set. In particular, the revised paper would benefit from citing a wider group of recent cross-domain studies that make the transferability argument concrete rather than symbolic. I do not think this needs to become an extensive literature review, but it should be more than a short list of three or four examples.

Comment 13: Page 26–27: I do not think the current revision fully addresses the broader-positioning point, and I encourage the authors to strengthen this part more deliberately. At present, the added discussion still reads as a brief acknowledgment that similar methods exist elsewhere, rather than a persuasive explanation of why the methodological logic of this paper matters beyond lung transplantation. This is important because one of the manuscript’s real strengths is not only its specific clinical application, but its potential relevance to a wider class of problems involving high-dimensional, structured, time-evolving data, limited sample sizes, nonlinear latent organization, and the need for interpretable prediction in high-stakes settings. In that sense, the paper would be much better positioned if the authors explicitly framed their contribution not simply as a transplant-risk model, but as an example of how topological feature engineering can enrich predictive learning when conventional representations may miss geometry, trajectory structure, or clinically meaningful shape information. That broader framing would give the work stronger interdisciplinary value and make the novelty feel more durable.

I therefore strongly encourage the authors to expand this paragraph into a more cohesive discussion supported by a richer and better-balanced citation set. In particular, the current revision would benefit from citing a wider range of recent studies showing that similar representation-learning challenges arise across physiological signal analysis, scientific imaging, geoscience, porous-media modeling, wireless engineering, and other compact scientific-ML domains. Examples that would fit naturally here include the ECG-focused survey by Ansari et al. in Artificial Intelligence Review (2025), which would help connect the manuscript’s signal-structure perspective to physiological diagnostics; the paleontology survey by Yaqoob et al. in Artificial Intelligence Review (2025), which shows how sparse and highly structured scientific patterns can be learned meaningfully outside medicine; “MicroCrystalNet” (IEEE Access, 2025), which is relevant as a lightweight and explainable architecture for microstructural scientific data; “PoreViT: Automated pore typing in carbonate rocks using vision transformers and neighborhood features” (Computers & Geosciences, 2025), which similarly addresses structured spatial representations in scientific classification; “FluidNet-Lite: Lightweight convolutional neural network for pore-scale modeling of multiphase flow in heterogeneous porous media” (Advances in Water Resources, 2025), which is a strong example of compact surrogate modeling in complex scientific systems; and “Prediction of Indoor Wireless Coverage from 3D Floor Plans Using Deep Convolutional Neural Networks” (Proc. IEEE LCN, 2021), which demonstrates how learned representations can replace or complement computationally expensive modeling in engineering prediction tasks. I would also encourage the authors to add further external references beyond this immediate citation circle, for example work on physics-informed machine learning for building-performance simulation and work on mobile-network coverage prediction using multimodal deep learning and semantic segmentation, because these would help make the broader applicability argument look balanced rather than self-contained.

My point here is not merely that the manuscript should cite more papers. Rather, I think the authors should use these references to argue more convincingly that the paper’s core methodological value lies in showing how topology-aware representations can support prediction in data-limited, high-stakes domains where ordinary tabular summaries may fail to capture latent structure. That is a much stronger and more compelling position than presenting the work as a narrowly specialized transplant application. If the authors develop this argument properly, the paper will not only read as more mature and better situated in the literature, but will also have clearer appeal to a broader scientific and engineering audience.

Comment 14: Page 27: The authors claim “This study introduces a topological attention-based predictive model for post-transplantation mortality.” I find “topological attention” still slightly imprecise and potentially confusing, because the method does not appear to implement an attention mechanism in the neural-network sense either. I encourage the authors to consider whether “topological feature engineering” or “topological feature-based prediction model” would be the cleanest and most transparent terminology, especially in the conclusion where readers look for a final statement of what was actually built.

Comment 15: Page 27 and Data Availability: I appreciate that the authors acted quickly and responsibly in removing the original repository and replacing it with synthetic data and documentation after the privacy concern was raised. This was the correct response. For completeness, I encourage the authors to ensure that the final manuscript’s Data Availability Statement fully matches the revised privacy-preserving access model and does not contain any residual wording suggesting unrestricted public access to real patient data.

Comment 16: Throughout the manuscript: The writing is improved, but I still encourage one last careful editorial pass for concision and terminology consistency. Most of the major clarity issues have been fixed, yet a few sections still read more like responses to reviewers than polished final prose. The manuscript is now much closer to publishable form, and a final tightening of phrasing, especially around novelty, timing, and broader applicability, would help it read as a cohesive final article rather than a revised draft.

7. PLOS authors have the option to publish the peer review history of their article (what does this mean?). If published, this will include your full peer review and any attached files.). If published, this will include your full peer review and any attached files.

**Do you want your identity to be public for this peer review?** For information about this choice, including consent withdrawal, please see our Privacy Policy..

Reviewer #1: None

Reviewer #3: No

  **Figure resubmission:** While revising your submission, we strongly recommend that you use PLOS’s NAAS tool (https://ngplosjournals.pagemajik.ai/artanalysis) to test your figure files. NAAS can convert your figure files to the TIFF file type and meet basic requirements (such as print size, resolution), or provide you with a report on issues that do not meet our requirements and that NAAS cannot fix.

After uploading your figures to PLOS’s NAAS tool - https://ngplosjournals.pagemajik.ai/artanalysis, NAAS will process the files provided and display the results in the "Uploaded Files" section of the page as the processing is complete. If the uploaded figures meet our requirements (or NAAS is able to fix the files to meet our requirements), the figure will be marked as "fixed" above. If NAAS is unable to fix the files, a red "failed" label will appear above. When NAAS has confirmed that the figure files meet our requirements, please download the file via the download option, and include these NAAS processed figure files when submitting your revised manuscript. **Reproducibility:** To enhance the reproducibility of your results, we recommend that authors of applicable studies deposit laboratory protocols in protocols.io, where a protocol can be assigned its own identifier (DOI) such that it can be cited independently in the future. Additionally, PLOS ONE offers an option to publish peer-reviewed clinical study protocols. Read more information on sharing protocols at https://plos.org/protocols?utm_medium=editorial-email&utm_source=authorletters&utm_campaign=protocols

---

## [Decision Letter · Decision Letter 2]

1 Apr 2026

Early Identification of High-Risk Individuals for Mortality after Lung Transplantation: A Retrospective Cohort Study with Topological Feature Engineering

PDIG-D-25-00813R2

Dear Dr Morilla,

We are pleased to inform you that your manuscript 'Early Identification of High-Risk Individuals for Mortality after Lung Transplantation: A Retrospective Cohort Study with Topological Feature Engineering' has been provisionally accepted for publication in PLOS Digital Health.

Best regards,

Vinod Kumar Chauhan, Ph.D.

Academic Editor

PLOS Digital Health

**Additional Editor Comments (if provided):**

**Reviewer Comments (if any, and for reference):**

Reviewer's Responses to Questions

**Comments to the Author**

1. If the authors have adequately addressed your comments raised in a previous round of review and you feel that this manuscript is now acceptable for publication, you may indicate that here to bypass the “Comments to the Author” section, enter your conflict of interest statement in the “Confidential to Editor” section, and submit your "Accept" recommendation.

Reviewer #3: All comments have been addressed

2. Does this manuscript meet PLOS Digital Health’s publication criteria? Is the manuscript technically sound, and do the data support the conclusions? The manuscript must describe methodologically and ethically rigorous research with conclusions that are appropriately drawn based on the data presented.? Is the manuscript technically sound, and do the data support the conclusions? The manuscript must describe methodologically and ethically rigorous research with conclusions that are appropriately drawn based on the data presented.

Reviewer #3: Partly

3. Has the statistical analysis been performed appropriately and rigorously?

Reviewer #3: Yes

4. Have the authors made all data underlying the findings in their manuscript fully available (please refer to the Data Availability Statement at the start of the manuscript PDF file)?

The PLOS Data policy requires authors to make all data underlying the findings described in their manuscript fully available without restriction, with rare exception. The data should be provided as part of the manuscript or its supporting information, or deposited to a public repository. For example, in addition to summary statistics, the data points behind means, medians and variance measures should be available. If there are restrictions on publicly sharing data—e.g. participant privacy or use of data from a third party—those must be specified.requires authors to make all data underlying the findings described in their manuscript fully available without restriction, with rare exception. The data should be provided as part of the manuscript or its supporting information, or deposited to a public repository. For example, in addition to summary statistics, the data points behind means, medians and variance measures should be available. If there are restrictions on publicly sharing data—e.g. participant privacy or use of data from a third party—those must be specified.

Reviewer #3: Yes

5. Is the manuscript presented in an intelligible fashion and written in standard English?

Reviewer #3: Yes

6. Review Comments to the Author

Reviewer #3: Thank you for addressing the comments. The paper is in a publishable state.

I wish you luck on your future researches!

7. PLOS authors have the option to publish the peer review history of their article (what does this mean?). If published, this will include your full peer review and any attached files.). If published, this will include your full peer review and any attached files.

**Do you want your identity to be public for this peer review?** For information about this choice, including consent withdrawal, please see our Privacy Policy..

Reviewer #3: No
